# Glycolysis inhibition induces anti-tumor central memory CD8⁺T cell differentiation upon combination with microwave ablation therapy

Xinyu Tang [1,2,9], Xinrui Mao[1,2,9], Peiwen Ling[1,2,9], Muxin Yu[1,2,9], Hua Pan[3,9], Jiaming Wang[1,2], Mingduo Liu[1,2], Hong Pan[1,2], Wen Qiu[4], Nan Che [5], Kai Zhang[2,6,7], Feifan Bao[8], Hongwei Peng[3], Qiang Ding [1,2], Shui Wang [1,2] ✉ & Wenbin Zhou [1,2] ✉

Minimally invasive thermal therapy is a successful alternative treatment to surgery in solid tumors with high complete ablation rates, however, tumor recurrence remains a concern. Central memory CD8⁺ T cells (T_CM) play important roles in protection from chronic infection and cancer. Here we find, by single-cell RNA analysis of human breast cancer samples, that although the memory phenotype of peripheral CD8⁺ T cells increases slightly after microwave ablation (MWA), the metabolism of peripheral CD8⁺ T cells remains unfavorable for memory phenotype. In mouse models, glycolysis inhibition by 2-deoxy-D-glucose (2DG) in combination with MWA results in long-term anti-tumor effect via enhancing differentiation of tumor-specific CD44^hiCD62L⁺CD8⁺ T_CM cells. Enhancement of CD8⁺ T_CM cell differentiation determined by Stat-1, is dependent on the tumor-draining lymph nodes (TDLN) but takes place in peripheral blood, with metabolic remodeling of CD8⁺ T cells lasting the entire course of the the combination therapy. Importantly, in-vitro glycolysis inhibition in peripheral CD8⁺ T cells of patients with breast or liver tumors having been treated with MWA thrice leads to their differentiation into CD8⁺ T_CM cells. Our work thus offers a potential strategy to avoid tumor recurrence following MWA therapy and lays down the proof-of-principle for future clinical trials.

Surgery is of great importance in the management of malignant tumors. Definitive resection only or combined with systemic therapy is curative in several parts of patients. However, postoperative recurrence occurs in up to a third of patients and carries a high risk of mortality after standard treatments[1]. Tumor recurrence or metastasis after surgery may be attributed to postoperative inflammation and remaining tumor cells, etc. Postoperative inflammation suppresses anti-tumor immunity, including upregulating adhesion molecules in target organs, recruiting immune cells capable of entrapping tumor cells[2], etc. Remnant tumor cells, such as circulating tumor cells disseminating from primary tumors, are promoted by postoperative inflammation to grow in distant organs[2,3]. Thus, there is an urgent need to find an effective therapy to improve the therapeutic effect further.

Minimally invasive thermal therapies, including microwave ablation (MWA), radiofrequency ablation, and cryoablation, have been gradually applied to treat solid tumors as standard local therapy,

including hepatocellular carcinoma, lung cancer, renal carcinoma, thyroid tumor, and so on[4–8]. Several clinical trials have confirmed the local effect of ablation therapies with high complete ablation rates in the treatment of breast cancer[9–11]. Recently, the long-term effect of local ablation therapies for early-stage breast cancer has also been reported[12–15].

Differently from surgery, thermal ablation of tumors can trigger anti-tumor immunity, which may be attributed to the tumor antigen or pro-inflammatory cytokine released from the ablated zone[15–17]. Given the suppressed anti-tumor immunity induced by surgery, thermal ablation may prolong patients' lifespans. Considering the local and systemic effects, thermal ablation may be a promising alternative to surgery for local therapy of solid tumors. However, the immune responses induced by local ablation are weak[15].

Robust CD8+T cell memory is essential for long-term protective immunity but is often compromised in cancer[18]. Memory CD8+T cells can be generally divided into effector memory cells (CD8+$T_{EM}$), central memory cells (CD8+$T_{CM}$), and tissue-resident memory cells (CD8+$T_{RM}$)[19]. Long-lived $T_{CM}$ cells can provide central immuno-surveillance by patrolling the lymphoid organs' draining peripheral tissue sites[20,21]. T cells with memory phenotypes such as $T_{CM}$ cells have proven superior to $T_{EM}$ or $T_{EFF}$ (effector) cells in mounting immune responses[22–25]. Inducing differentiation of memory T cells, such as $T_{CM}$ cells, becomes a reliable way to improve the long-term anti-tumor effect[26].

The metabolism of T cells regulates their fate, function, and longevity[25,27–30]. Metabolism modulation of T cells has been recognized as a potential therapeutic target to enhance or suppress immune responses in numerous settings, including anti-tumor immunity. As quiescent cells, memory T cells preferentially rely on oxidative phosphorylation (OXPHOS) relative to aerobic glycolysis. In contrast, the upregulation of aerobic glycolysis is regarded as a hallmark of T-cell activation. Differentiation of memory CD8+T cells and anti-tumor function of CD8+T cells have proven to be promoted by interfering with glycolysis[31], which may be a promising strategy to enhance the long-term memory phenotype of ablation therapy.

Here we report that the memory phenotype of the peripheral CD8+T cells is mildly enhanced after MWA of human breast cancer; however, the metabolic status of the peripheral CD8+T cells relies on glycolysis more than on OXPHOS, which is unfavorable for the generation of memory CD8+T cells. In multiple tumor models, gly-colysis inhibition after MWA enhances the systemic long-term anti-tumor effect via promoting CD8+$T_{CM}$ cell differentiation. Mechanically, promoting CD8+$T_{CM}$ cell differentiation is tumor-draining lymph node (TDLN) dependent but takes place in the peripheral blood. Notably, the peripheral CD8+T cells from patients with breast or liver tumors after MWA are promoted to differentiate into CD8+$T_{CM}$ cells by inhibiting glycolysis in vitro. Glycolysis inhibition after MWA may be a promising treatment strategy for early-stage malignant tumors.

## Results
### Memory and metabolic features of peripheral CD8+T cells in patients with breast cancer post-local ablation
The landscape of the peripheral immune response induced by local MWA in breast cancer patients has been reported using single-cell RNA sequencing[17]. The peripheral blood of six patients from the clinical trial (ChiCTR2000029155) was collected before and 1 week after MWA (Fig. 1a). By analyzing the single-cell RNA sequencing data, we reported the memory and metabolic features of the peripheral CD8+T cells after MWA.

To determine the memory features of CD8+T cells, a memory marker IL7R and two memory CD8+T cell signatures[32,33] were applied in this study. Of the CD8+T cells, two cell clusters were identified, including GZMH+CD8+T cells and GZMK+CD8+T cells (Fig. 1b, c,

Supplementary Fig. 1a). IL7R was then analyzed in both the CD8+T-cell clusters (Supplementary Fig. 1b). Compared to that before MWA, the expression level of IL7R was significantly higher after ablation in the GZMK+CD8+T cell cluster (Fig. 1d). In the GZMH+CD8+T cell cluster, IL7R expression level didn't increase with a significant difference after MWA (Fig. 1d). To further investigate the memory phenotype induced by MWA, we compared the proportion of IL7R+CD8+T cells after abla-tion with that before MWA. The proportion of IL7R+CD8+T cells increased after ablation but without significant differences (Fig. 1e). Consistent with IL7R, the CD8+T cell memory scores showed similar results (Fig 1f[32], Supplementary Fig. 1c[33]). Thus, the memory phenotype slightly increased after MWA.

Unlike effector T cells, memory T cells preferentially rely on OXPHOS relative to glycolysis, both of which are important energy sources of T cells. To explore the metabolic changes, especially energy metabolism changes in the peripheral CD8+T cells after ablation ther-apy, Gene Set Variation Analysis (GSVA) algorithm was utilized to cal-culate the metabolic scores of metabolism pathways from the KEGG dataset (Fig. 1g). The scores of OXPHOS decreased in both clusters of the CD8+T cells after ablation, while those of glycolysis showed no difference (Fig. 1g–h). To further clarify which energy metabolism pathway the peripheral CD8+T cells mainly relied on after ablation, delta score (Δscore) was used, defined as the D-value between glyco-lysis and OXPHOS. The Δscore increased significantly after MWA in both the GZMH+CD8+T cell cluster and GZMK+CD8+T cell cluster (Fig. 1i), suggesting more contribution made by glycolysis than that by OXPHOS to the energy supply of the CD8+T cells after ablation. After ablation, the metabolism status of the CD8+T cells might be unfavor-able for the generation of long-lived memory CD8+T cells.

### Interfering with glycolysis potentiates the anti-tumor effect of complete local ablation
The metabolic features of the CD8+T cells after ablation were unfa-vorable for memory CD8+T cell differentiation. Interfering with gly-colysis could induce CD8+T cells to preferentially use OXPHOS instead of glycolysis as an energy source, strengthening memory CD8+T cell differentiation[31]. We hypothesized that interfering with glycolysis after MWA of solid tumors can promote memory CD8+T cell differentiation, providing an enhanced long-term anti-tumor effect.

A BALB/c mouse-derived transplantable 4T1 breast tumor model was applied. 2-deoxy-D-glucose (2DG), which can block glycolysis through hexokinase and phosphoglucose isomerase inhibition[34], was used to interfere with glycolysis in this study. To simulate the clinical management of early-stage malignant tumors, the primary 4T1 tumor without apparent metastases was treated 10 days after the inoculation (Fig. 2a). Different from previously reported partial ablation researches[35–37], the primary tumor of the animal model in our study was completely ablated. All mice were randomly divided into four groups to compare the efficacy of different therapies. Consistent with our human observation, the glycolysis bias of CD8 + T cells in the mouse model after MWA was confirmed (Supplementary Fig. 2i). The TDLNs and lungs of mice were harvested on day 35 for pathological examination (Fig. 2a). The TDLNs of mice from the 2DG and PBS groups showed larger volumes than those from the MWA + 2DG and MWA + PBS groups (Fig. 2b, Supplementary Fig. 2a). No metastasis was found in the TDLNs from the MWA + 2DG and MWA + PBS groups (Fig. 2b), but metastases were found in the 2DG and PBS groups. Moreover, the mice in the MWA + 2DG and MWA + PBS groups exhib-ited complete prevention of metastatic lesions in the lungs. In contrast, apparent lung metastases were observed in the 2DG and PBS groups (Fig. 2b, Supplementary Fig. 2b, c). The results confirmed the ther-apeutic effect of MWA.

To further clarify the immunologic nature of the combination therapy, we re-implanted the mice from four groups with 4T1 tumor cells at the contralateral flank 5 days after the last injection of 2DG,

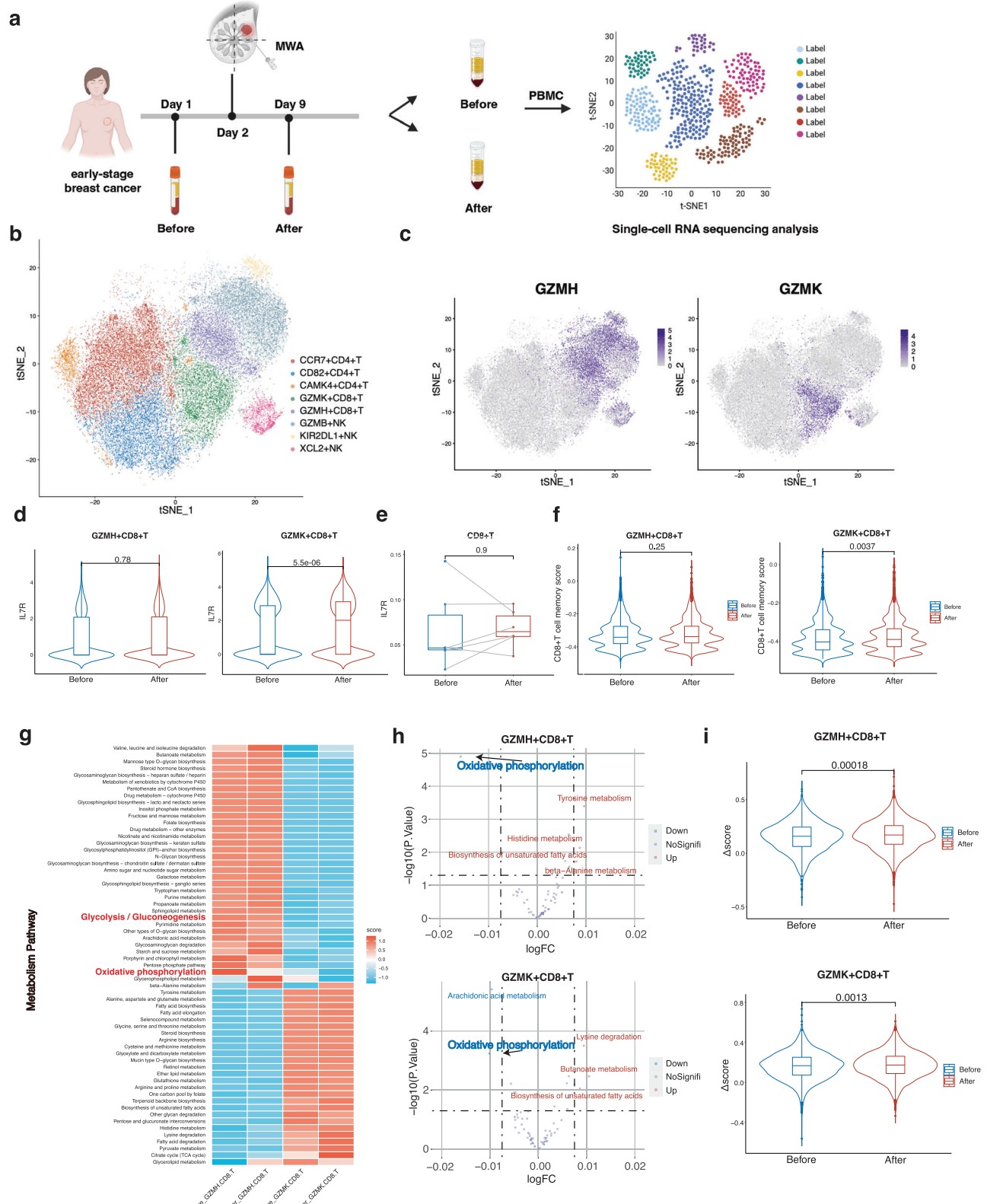

eliminating the direct anti-tumor effect of the remaining 2DG with a plasma half-life of only 48 min[38] (Fig. 2c, d). Also, no 2DG was detected 5 days after the last injection of 2DG (Fig. 2c, d). Even though the tumor cells were re-challenged with a high number of $5 \times 10^5$, two of five mice after the combination therapy completely eradicated a second tumor implantation. In contrast, a second tumor was observed at the opposite flank of the mice from the other three groups (Fig. 2e,

Supplementary Fig. 2d). Moreover, the re-challenged tumor growth was significantly delayed in the MWA + 2DG group compared to the other three groups (Fig. 2e, Supplementary Fig. 2d). The experiment was replicated in another mammary tumor model Py8119 and mice in the 2DG and PBS were executed to adhere to animal ethics before day 35. Similar results were obtained (Fig. 2f, Supplementary Fig. 2e, j). Notably, the survival of mice after re-challenge in the MWA + 2DG

**Fig. 1 | Memory and metabolic features of peripheral CD8⁺T cells in patients with breast cancer post-local ablation. a** Experimental design for (**b–i**). Peripheral blood of six patients with breast cancer was collected before and 1 week after MWA. Peripheral blood mononuclear cells (PBMC) were isolated for single-cell RNA sequencing. **b** T-distributed stochastic neighbor embedding (t-SNE) plot of eight distinct clusters demarcated by colors based on gene expression differences in NK and T cells. **c** Feature plots of key gene expression in the CD8⁺T cell clusters. **d** Violin plots of differential expression of known memory T cell markers (IL7R) in two CD8⁺T cell clusters. **e** Box plots of the proportions of IL7R⁺CD8⁺T cells in peripheral CD8⁺T cells before and after ablation ($n = 6$). **f** Violin plots of scores of memory CD8⁺T cells in two CD8⁺T cell clusters. **g** Heatmap of the average metabolic pathway scores of two CD8⁺T cell clusters before and after ablation. The metabolic pathways highlighted in red represent the main energy metabolism pathways. The

metabolic pathway scores were determined by the GSVA algorithm. **h** Volcano plots of differential metabolic pathways of two CD8⁺T cell clusters before and after MWA of breast cancer. **i** Violin Plots of Δscore (glycolysis - OXPHOS) of two CD8⁺T cell clusters before and after ablation. Significance was determined as $P < 0.05$. The low ends of the segment indicate the minimum and the high ends of the segment indicate the maximum. Lower bounds of the box indicate the 25th percentile and the higher bounds of the box indicate the 75th percentile. The segment in the middle is the median (**d–f, i**). Significance determined by two-tailed unpaired t-test (**d, f, i**), two-tailed paired t-test (**e**), and two-sided linear model fitting and empirical Bayesian methods (**h**). Source data are provided as a Source Data file. **a** Created with BioRender.com released under a Creative Commons Attribution-NonCommercial-NoDerivs 4.0 International license.

group was significantly prolonged than that in the other groups (Fig. 2g). The abscopal effect confirmed the anti-tumor effect of the combination therapy.

Tumors with different immune infiltrations and immunogenicities show fundamentally different treatment responses. Thus, it was crucial to assess whether the combination therapy could induce a stronger anti-tumor effect in other types of tumors. C57BL/6 mouse-derived colon adenocarcinoma cell line MC38 and melanoma cell line B16F10 were applied to replicate our re-challenge experiment (Fig. 2h). The mice subcutaneously implanted with B16F10 or MC38 tumor cells were randomly divided into four groups, as described above. For the MC38 model, all mice in the MWA + 2DG group eradicated tumor cells re-challenge and three of four mice in the MWA + PBS group eradicated tumor cells re-challenge (Fig. 2j, Supplementary Fig. 2g). However, a second tumor was observed in all mice in the 2DG and PBS groups (Fig. 2j, Supplementary Fig. 2g). Tumor growth of mice in the MWA + 2DG and MWA + PBS groups was significantly delayed compared with the PBS group (Fig. 2j, Supplementary Fig. 2g). For the B16F10 model, as mice in the 2DG and PBS groups were executed on day 15, the tumor re-challenge was then performed on the left two groups on day 35. Two mice in the MWA + 2DG group eliminated tumor cell re-challenge but not in the MWA + PBS group (Fig. 2i, Supplementary Fig. 2f). Tumor growth of mice in the MWA + 2DG group was significantly delayed compared with the MWA + PBS group (Fig. 2i, Supplementary Fig. 2f). Together, glycolysis inhibition after MWA induced a strong systemic anti-tumor effect in several types of tumors regardless of the totally different immunologic features.

After the above four different treatments of the 4T1 tumor, several mice in the MWA + 2DG group survived for a very long time, while no mice in the other groups survived longer than 140 days. To explore the very long-term anti-tumor effect of the combination therapy, three surviving mice after the combination therapy were re-challenged with 4T1 tumor cells on day 140, and three naïve mice were enrolled as control (Fig. 2k). Interestingly, all mice in the MWA + 2DG group eradicated 4T1 tumor cells while all mice in the naïve group did not (Fig. 2l, Supplementary Fig. 2h). On day 15 after the re-challenge, the volumes of re-implanted tumors in the naïve group were larger than those in the MWA + 2DG group (Fig. 2l, Supplementary Fig. 2h). The results indicated that the combination therapy induced a relatively long-term anti-tumor effect.

**Interfering with glycolysis post local ablation induces enhanced long-term memory phenotype of peripheral CD8⁺T cells**

The elimination of tumor cell re-challenge might be attributable to systemic anti-tumor immunity. To clarify the immune response in the groups described above, we assessed T cells and their subsets in the spleens and peripheral blood on day 35 (Fig. 3a, Supplementary Fig. 3a). For mice implanted with 4T1 mammary tumor cells, significantly increased percentages and absolute numbers of CD3⁺T cells, CD4⁺T cells, and CD8⁺T cells were observed in the MWA + 2DG and MWA + PBS groups compared to the 2DG and PBS groups (Fig. 3b).

Meanwhile, the mice from the MWA + 2DG group showed higher percentages of CD3⁺T and CD8⁺T cells compared to those from the MWA + PBS group (Fig. 3b).

To further investigate the subsets of CD4⁺T and CD8⁺T cells, activation marker CD44 and naïve/memory marker CD62L were tested (Fig. 3b, Supplementary Fig. 3a). CD44^hiCD62L⁻T cells are regarded as $T_{EM}/T_E$ (effector memory or effector), and CD44^hiCD62L⁺T cells are regarded as $T_{CM}$ cells (central memory). For mice implanted with 4T1 mammary tumor cells, higher percentages and absolute numbers of the four subsets of T cells were observed in the spleens of mice after MWA than those without MWA (Fig. 3b). In addition, the percentage and absolute number of CD44^hiCD62L⁺CD8⁺T cells in the spleens of mice from the MWA + 2DG group was significantly higher than that in the MWA + PBS group, but not other subsets of T cells (Fig. 3b). Similar results were obtained in another mammary tumor model Py8119 (Fig. 3c). Moreover, the main results were observed in the peripheral blood (Supplementary Fig. 3b–e).

As previously reported, glycolysis inhibition could promote T cells to preferentially use OXPHOS instead of glycolysis as an energy source, which might induce more Tregs[39]. No difference in percentages of Tregs in the spleens of mice between the MWA + 2DG and MWA + PBS groups was observed (Fig. 3f). As one of the essential energy sources of CD8⁺T cells, glycolysis deprivation might also induce apoptosis of CD8⁺T cells after MWA. No significant difference in the non-apoptotic CD44^hiCD62L⁺CD8⁺T cells was found in the spleens of mice between the MWA + 2DG and MWA + PBS groups (Fig. 3g). Although 2DG can directly inhibit the growth of tumor cells, glycolysis inhibition might also negatively influence the activation of CD8⁺T cells when they are exposed to the cognate antigens. To test the overall anti-tumor effect of MWA + 2DG compared with MWA + PBS during the memory induction phase, CD8⁺T cells were isolated from spleens of mice in the MWA + 2DG and MWA + PBS groups 7 days after MWA to coculture with 4T1 tumor cells in the presence of 2DG or PBS respectively (Fig. 3h). No significant difference in the overall anti-tumor response between the 2 groups was found (Fig. 3j). The above results indicated that the combination therapy didn't influence the short-term overall anti-tumor effect, but significantly increased percentages and numbers of long-lived CD44^hiCD62L⁺CD8⁺T cells.

In order to better understand the long-term immune effect of the combination therapy, we assessed the percentages and absolute numbers of memory T cells in the surviving mice in the MWA + 2DG group. The spleens of mice implanted with 4T1 mammary tumor cells from the MWA + 2DG group were obtained on day 35 and day 140 (Fig. 3k). The percentage and absolute number of CD44^hiCD62L⁺CD8⁺T cells on day 140 were significantly higher than that on day 35 (Fig. 3m). Of CD44^hiCD8⁺T cells (activated CD8⁺T cells), the level of CD62L expression was also much higher on day 140 than that on day 35 (Fig. 3l). The results demonstrated that interfering with glycolysis post-local ablation induced an increasingly robust and long-lasting memory phenotype in peripheral CD8⁺T cells as time progressed.

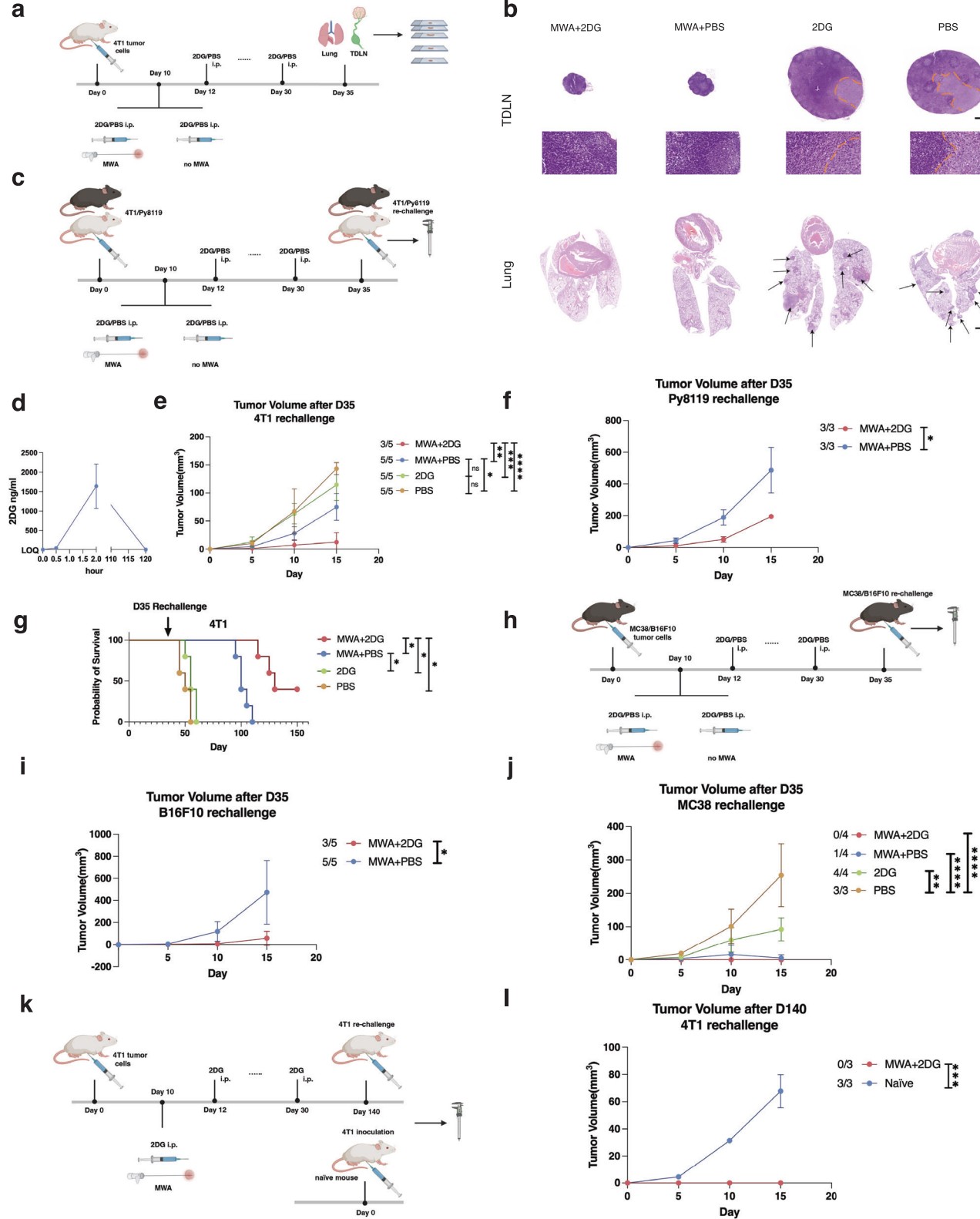

## Interfering with glycolysis boosts the anti-tumor effect of local ablation in a CD8[+]T cell-dependent manner

To determine whether T cells played a pivotal role in the different immune responses between the MWA + 2DG and MWA + PBS groups after tumor re-challenge, BALB/cAnSlac-nu mice without mature T cells were used in our study. All mice were randomly divided into the MWA + 2DG and MWA + PBS groups after inoculating 4T1 tumor cells subcutaneously (Fig. 4a). On day 35, 4T1

tumor cells were re-challenged as above to evaluate the anti-tumor effect (Fig. 4a). The deficiency of T cells resulted in the growth of tumors in both MWA + 2DG and MWA + PBS groups. Additionally, no significant difference in the tumor volume was observed between the two groups 15 days after the re-challenge (Fig. 4b, Supplementary Fig. 4a). Overall, T cells played a vital role in the long-term anti-tumor effect of the combination therapy.

**Fig. 2 | Interfering with glycolysis potentiates the anti-tumor effect of complete local ablation. a** Experimental design for (**b**). **b** Representative H&E stained sections of lung and TDLN metastases in (**a**) ($n = 4$ per group). Arrow, lung metastases. Scale bar, from top to bottom, 500 μm, 20 μm, and 1000 μm. **c** Experimental design for (**d, e, f**). Respectively, 4T1 or Py8119 tumor cells ($n = 5 \times 10^5$) were re-challenged at the opposite flanks of mice on day 35. **d** Concentration of 2DG in the plasma after the last 2DG injection detected by UPLC-ESI-MS/MS. **e** Tumor volume curves of mice in (**c**) re-challenged with 4T1 ($n = 5$ per group). **f** Tumor volume curves of mice in (**c**) re-challenged with Py8119 ($n = 3$ per group). **g** Survival curves of mice in (**d**) re-challenged with 4T1 ($n = 5$ per group). **h** Experimental design for (**i, j**). MC38 or B16F10 tumor cells ($n = 5 \times 10^5$) were re-challenged at the opposite flanks of mice on day 35, respectively. **i, j** Tumor volume curves of mice in (**h**) re-challenged with B16F10 or MC38 ($n = 5$ per group for the B16F10 model; $n = 4$ in the

MWA + 2DG, MWA, and 2DG groups and $n = 3$ in the NC group for the MC38 model). **k** Experimental design for (**l**). 3 mice in the MWA + 2DG group survived for 140 days and were re-challenged with 4T1 tumor cells ($n = 1 \times 10^5$) at the opposite flanks of mice from the 2 groups on day 140. Naïve BALB/c mice were set as control. **l** Tumor volume curves of mice in (**k**) re-challenged with 4T1 ($n = 3$ per group). All the experiments were repeated for at least two times. Data are mean ± SD (**d, e, f, i, j, l**). Significance determined by one-way ANOVA (**e, j**), two-tailed unpaired t-test (**f, i, l**), and log-rank (Mantel-Cox) test with Bonferroni's correction (**g**). ns not significant, *$P < 0.05$, **$P < 0.01$, ***$P < 0.001$, ****$P < 0.0001$. Exact p values and source data are provided as a Source Data file. **a, c, h, k** Created with BioRender.com released under a Creative Commons Attribution-NonCommercial-NoDerivs 4.0 International license.

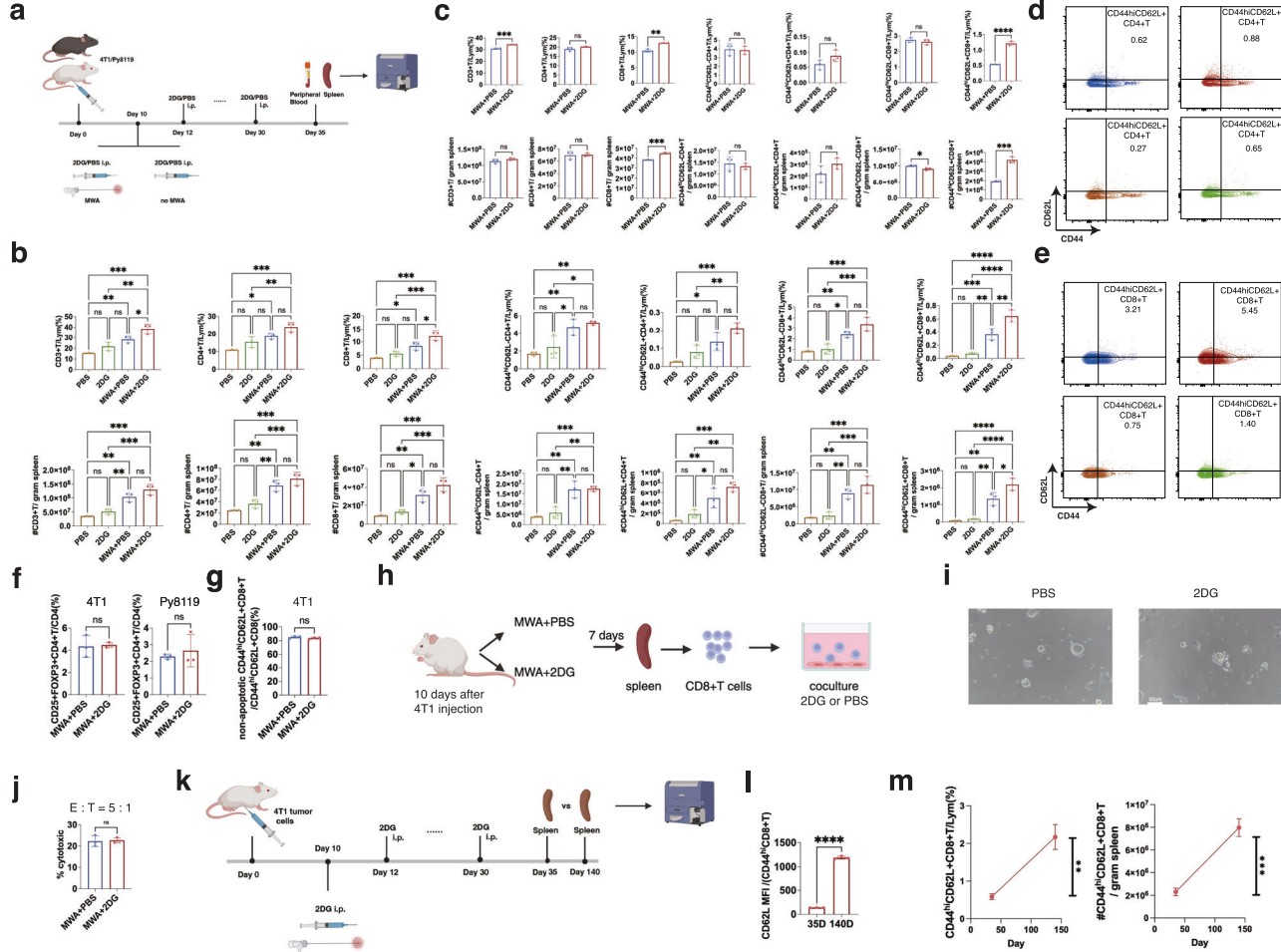

**Fig. 3 | Interfering with glycolysis post local ablation induces enhanced long-term memory phenotype of peripheral CD8⁺T cells. a** Experimental design for (**b-g**). Percentages and absolute numbers of CD3⁺T, CD4⁺T, CD8⁺T, CD44^hiCD62L⁻ CD4⁺T, CD44^hiCD62L⁺CD4⁺T, CD44^hiCD62L⁻CD8⁺T, CD44^hiCD62L⁺CD8⁺T cells in spleens of mice implanted with 4T1 (**b**, $n = 3$ per group) or with Py8119 (**c**, $n = 3$ per group). **d, e** Representative flow plots of CD44^hiCD62L⁺CD8⁺T and CD44^hiCD62L⁺ CD4⁺T cells in (**b**). **f, g** Percentages of Tregs and non-apoptotic memory CD8 + T cells in spleens of mice implanted with 4T1 ($n = 3$ per group). **h** Experimental design for (**i, j**). CD8 + T cells were isolated from the spleens of mice from MWA + 2DG and MWA + PBS on day 17 to coculture with 4T1 tumor cells in the presence of 2DG or PBS respectively for 24 h. Representative pictures of the coculture experiment

(**i**) and percentages of cytotoxicity in the coculture experiment (**j**, $n = 3$ per group). **k** Experimental design for (**l, m**). **l** MFI of CD62L of CD44^hiCD8⁺T cells between day 35 and day 140 ($n = 3$ per group). **m** Dynamic changes of CD44^hiCD62L⁺CD8⁺T cell percentage and absolute number in spleens of mice from the MWA + 2DG group as time progressed ($n = 3$ per group). All the experiments were repeated for at least two times. Data are mean ± SD (**b, c, f, g, j, l, m**). Significance determined by one-way ANOVA (**b**) and two-tailed unpaired t-test (**c, f, g, j, l, m**). ns, not significant, *$P < 0.05$, **$P < 0.01$, ***$P < 0.001$, ****$P < 0.0001$. Exact p values and source data are provided as a Source Data file. **a, h, k** Created with BioRender.com released under a Creative Commons Attribution-NonCommercial-NoDerivs 4.0 International license.

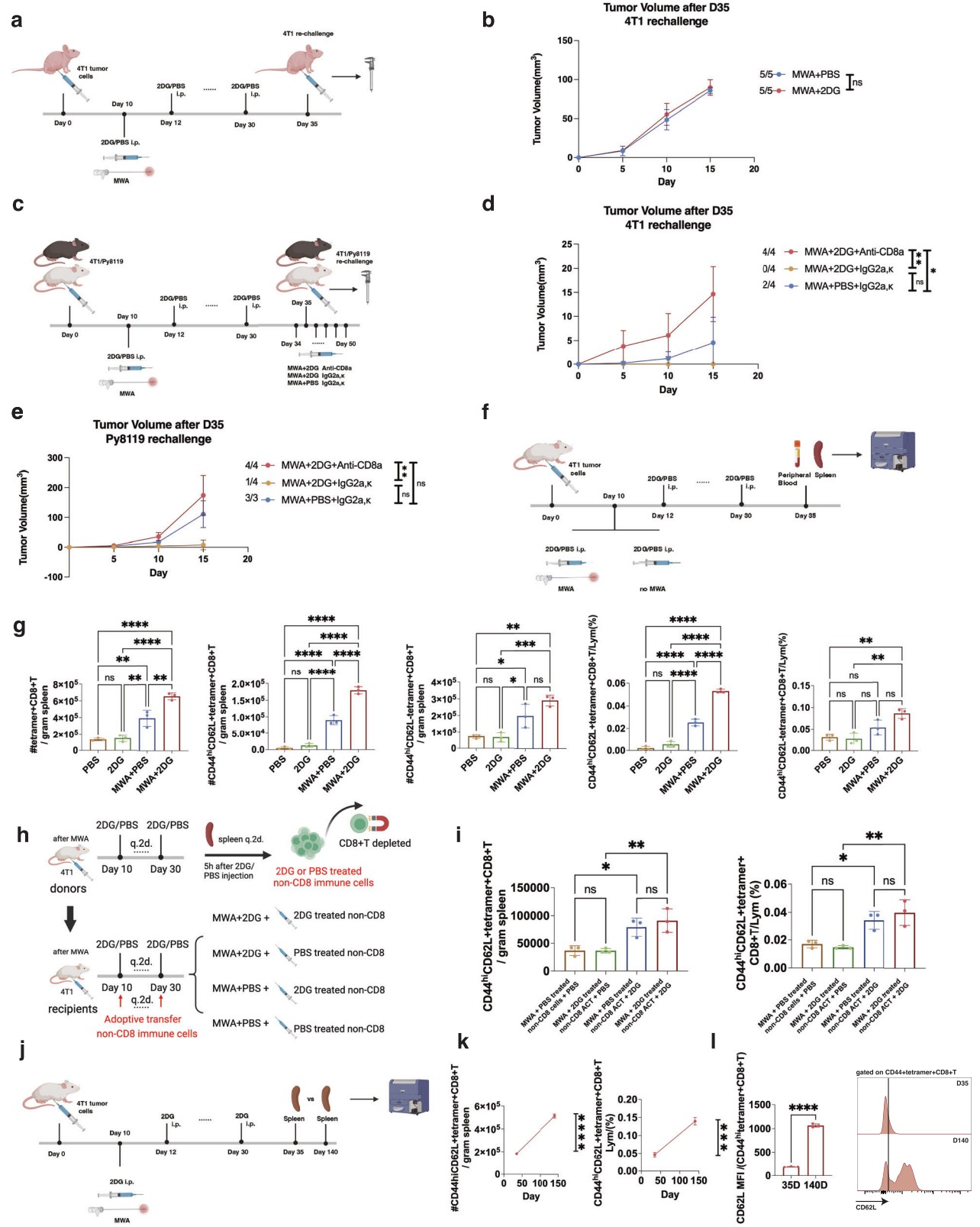

CD8+T cells are the most potent killers in T-cell mediated immune effect. To validate the contribution of CD8+T cells, we depleted CD8+T cells or not in mice after the combination therapy, and mice treated with MWA alone were set as control (Fig. 4c). 4T1 or Py8119 tumor cells were re-challenged as described above (Fig. 4c). For 4T1 models, MWA exhibited only a slightly enhanced memory phenotype with the complete elimination of 4T1 tumor in two mice among the

three groups (Fig. 4d). In the groups with or without CD8+T depletion after the combination therapy, all mice injected with IgG control antibody eradicated a tumor, while those with depletion of CD8+T cells did not (Fig. 4d, Supplementary Fig. 4b). Obviously, the tumor volumes were larger after the depletion of CD8+T cells than those not (Fig. 4d). The main results were also observed in another mammary tumor model Py8119 (Fig. 4e, Supplementary Fig. 4c). These results

**Fig. 4 | Interfering with glycolysis boosts the anti-tumor effect of local ablation in a CD8⁺T cell-dependent manner. a** Experimental design for (**b**). 4T1 tumor cells ($n = 1 \times 10^5$) were rechallenged at the opposite flanks of mice on day 35. **b** Tumor volume curves of the mice in (**a**) re-challenged with 4T1 (n = 5 per group). **c** Experimental design for (**d**, **e**). Mice intraperitoneally injected with anti-CD8a or rat IgG2a,κ q.4d from day 34 to day 50. 4T1 tumor cells ($n = 1 \times 10^5$) were re-challenged at the opposite flanks of mice on day 35. **d, e** Tumor volume curves of the mice (*n* = 4 per group for 4T1 model; *n* = 4 for MWA + 2DG with anti-CD8a or IgG2a,κ group and *n* = 3 for MWA + PBS with IgG2a,κ group for Py8119 model). **f** Experimental design for (**g**). **g** Percentages and absolute numbers of tetramer⁺CD8⁺T, CD44^hiCD62L⁻tetramer⁺CD8⁺T, and CD44^hiCD62L⁺tetramer⁺CD8⁺T cells in spleens of mice (*n* = 3 per group). **h** Experimental design for (**i**). **i** Percentage and absolute number of

CD44^hiCD62L⁺tetramer⁺CD8⁺T cells in spleens of mice (*n* = 3 per group). **j** Experimental design for (**k, l**). **k** Dynamic changes of CD44^hiCD62L⁺tetramer⁺CD8⁺T cell percentage and absolute number in spleens of mice from the MWA + 2DG group as time progressed (*n* = 3 per group). **l** CD62L expression of CD44^hitetramer⁺CD8⁺ T cells in spleens of mice from the MWA + 2DG group as time progressed (*n* = 3 per group). All the experiments were repeated for at least two times. Data are mean ± SD (**b, d, e, g, i, k, l**). Significance determined by one-way ANOVA (**d, e, g, i**) and two-tailed unpaired t-test (**b, k, l**). ns, not significant,*$P < 0.05$, **$P < 0.01$, ***$P < 0.001$, ****$P < 0.0001$. Exact $p$ values and source data are provided as a Source Data file. **a, c, f, h, j** Created with BioRender.com released under a Creative Commons Attribution-NonCommercial-NoDerivs 4.0 International license.

supported that the control of re-challenged tumors in the MWA + 2DG group relied on CD8⁺T cells.

To exclude the influence of bystander T cells, 4T1 tumor antigen AH1[40] specific tetramer was used to determine 4T1 tumor-specific T cell clone (Fig. 4f). The absolute number of tetramer⁺CD8⁺T cells was increased in the spleens of mice from the MWA + 2DG group compared to the other 3 groups (Fig. 4g). Consistently, gated on tumor-specific CD8⁺T cells, the percentages and absolute numbers of CD44^hiCD62L⁺CD8⁺T cells in spleens of mice were significantly higher than the other 3 groups but not the CD44^hiCD62L⁻CD8⁺T cells (Fig. 4g). The main results were also obtained in the peripheral blood (Supplementary Fig. 4d). Considering all the above results, MWA played an important role in the initial phase of the enhanced anti-tumor effect and immune responses of CD8⁺T cells induced by the combination therapy, which might be attributed to the antigen release effect of MWA[16]. After MWA, necrosis of tumor cells with the release of intracellular zsgreen into TDLN captured by dendritic cells (DC) was found compared to the baseline (Supplementary Figs. 4e–f and 5d), supporting the antigen release effect of MWA.

It was important to exclude the effect of 2DG on the non-CD8 immune cells which might contribute to inducing memory phenotype after the combination therapy based on our hypothesis that 2DG promoted the memory phenotype of CD8⁺T cells directly. Immune cells of donor mice treated with MWA + 2DG or MWA + PBS as above were collected 5 h after the injection of 2DG or PBS every 2 days. The isolated immune cells with CD8⁺T cells depleted were transferred into the recipient mice treated with MWA + 2DG or MWA + PBS as above (Fig. 4h). Percentage and absolute number of tumor-specific CD44^hiCD62L⁺CD8⁺T cells were not increased after transfer of 2DG treated non-CD8 immune cells compared to transfer of PBS treated non-CD8 immune cells in the mice treated with MWA + PBS (Fig. 4i). Thus, other immune cells had no significant impact on memory T cell development. The results indicated the direct effect of 2DG on CD8⁺T cells in the combination therapy.

The long-term memory phenotype of the tumor-specific CD8⁺T cells after the combination therapy should also be determined by excluding bystander T cell clones. The percentage and absolute number of tumor-specific CD44^hiCD62L⁺CD8⁺T cells in the spleens of mice from the combination group on day 140 was significantly higher than that on day 35 (Fig. 4j, k). Of tumor-specific CD44^hiCD8⁺T cells (activated CD8⁺T cells), the level of CD62L expression was also much higher on day 140 than that on day 35 (Fig. 4l). Overall, the results indicated the combination therapy induced robust tumor-specific memory CD8⁺T cell immune response systemically.

**TDLN plays a critical role in the generation of systemic memory phenotype of the combination therapy**
Lymph node is the classical site for immune response generation, and TDLN may be necessary for the abscopal effect of local therapy for tumors. We then focused on whether TDLN was crucial in the memory

immune response induced by the combination therapy. TDLN resection was performed during the ablation of the primary tumor on day 10, and 2DG was still used as before (Fig. 5a). Compared to the group with sham surgery of the TDLNs, the TDLN resection didn't change the percentages of CD8⁺T cells and CD4⁺T cells in the spleens (Fig. 5b). However, the percentages of CD8⁺T cells and CD4⁺T cells in the peripheral blood showed a significant decrease with TDLN resection (Fig. 5c). Notably, the percentages of CD44^hiCD62L⁺CD8⁺T cells in the spleens and peripheral blood in the TDLN resection group were significantly lower than those with sham surgery (Fig. 5d). FTY720 blocking lymphocytes egress from the lymph node was used or not during the combination therapy to further validate the importance of TDLN (Fig. 5e). For the 4T1 model, percentages and absolute numbers of peripheral CD8⁺T, tumor-specific CD8⁺T, and tumor-specific CD44^hiCD62L⁺CD8⁺T cells in the FTY720 group decreased significantly compared to those with 0.9% saline (Fig. 5f, h, i). Similar results were also obtained in the Py8119 model (Fig. 5g). These results suggested that TDLN played a critical role in the generation of systemic memory phenotypes induced by the combination therapy.

**Immune microenvironment in TDLN is more activated by the combination therapy compared with local ablation alone**
To further compare the intrinsic immunologic changes of TDLN between the MWA + 2DG and MWA + PBS groups, bulk RNA-seq of TDLN was performed on day 15 (Fig. 5j–l). The Cibersort algorithm was utilized to analyze the resident immune cells in the TDLNs. The percentage of CD8⁺T_naïve cells showed no difference between the MWA + 2DG and MWA + PBS groups, while that of CD8⁺T_M+E (memory and effector) during the combination therapy tended to increase compared to the MWA + PBS group (Fig. 5j). To figure out the changes in the immunologic process between the TDLNs in the MWA + 2DG and MWA + PBS groups, pathway enrichment analysis was used. GO (gene ontology) pathway enrichment analysis indicated that T cells were more activated by inhibiting glycolysis after MWA (Fig. 5k). The scores of pathways from the GO dataset which were related to CD8⁺T cells were analyzed by Gene Set Variation Analysis (GSVA). Consistently, the average scores of CD8⁺T cell activation and proliferation were higher with glycolysis inhibition after MWA (Fig. 5l). CD4⁺Th1 cells are known as helper T cells which mainly assist CD8⁺T cell activation. In our Cibersort results, the ratio of Th1 cells was significantly higher after the combination therapy (Fig. 5j). The average scores of Th1 cell differentiation and Th1 type immune response were higher in the MWA + 2 DG group than in the MWA + PBS group (Fig. 5l). Moreover, the pathways related to the functions of Th1 cells, such as IL2 and IFNγ production, showed higher scores in the MWA + 2DG group (Fig. 5l). The above results indicated that glycolysis inhibition might further activate CD8⁺T cells and promote the functions of Th1 cells in the TDLN after MWA. Consistently, Percentages of CXCR3⁺CD4⁺Th1, CD8⁺T, CD25⁺CD8⁺T, and CD69⁺CD8⁺T cells were elevated in the MWA + 2DG group compared to the MWA + PBS group (Fig. 5m, n, Supplementary Fig. 5a–c).

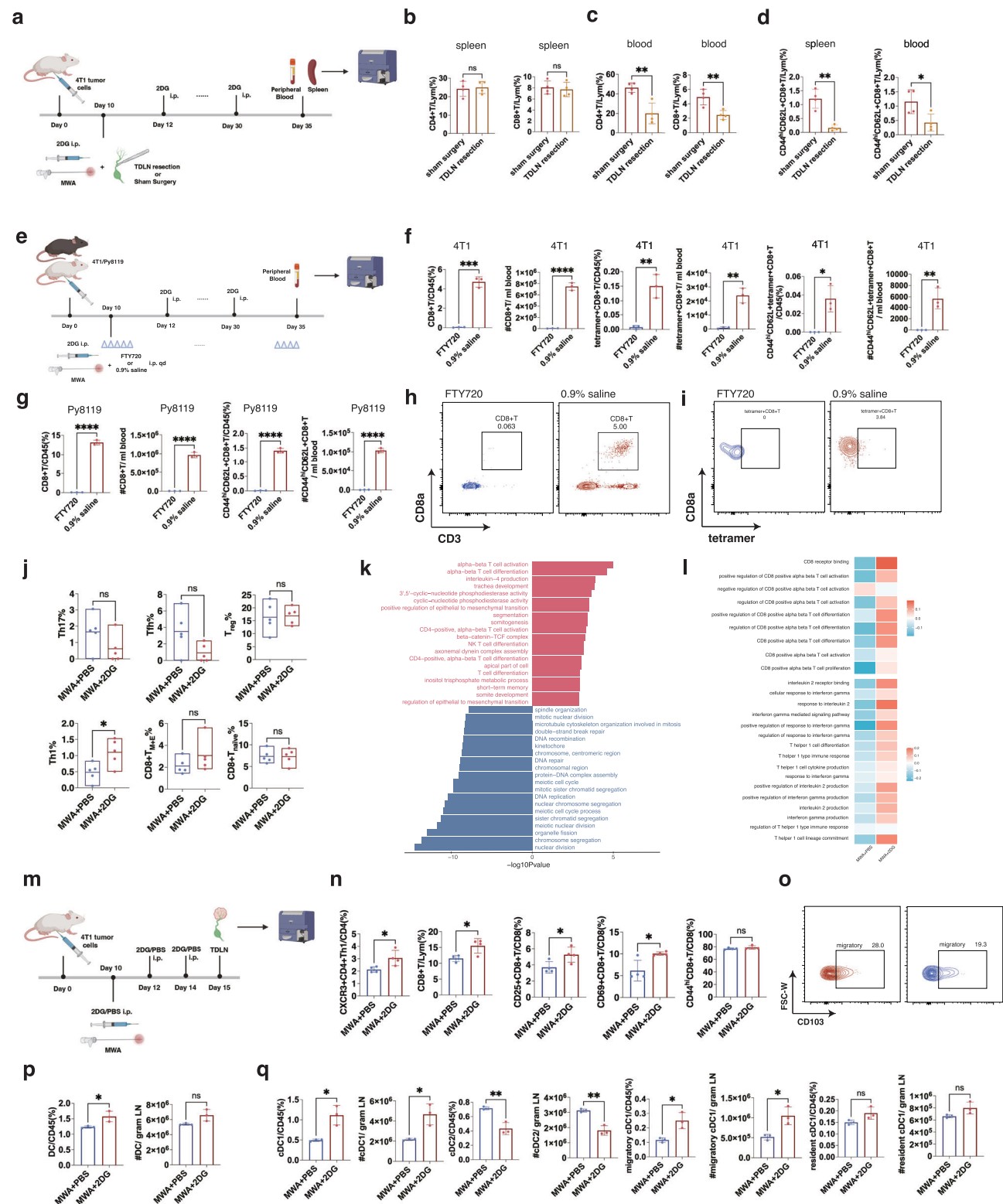

As the trigger for adaptive immune response, DCs were also evaluated. A higher percentage of DCs was induced by the combination therapy in the TDLNs compared to the MWA + PBS on day 15 (Fig. 5p, Supplementary Fig. 5d). Moreover, the percentages and absolute numbers of cDC1s especially migratory cDC1s were higher in the TDLNs of mice from the MWA + 2DG group than those in the MWA + PBS group (Fig. 5q). Overall, glycolysis inhibition after MWA further activated CD8+T cells with more Th1 cells and migratory cDC1s, which might be the potential upstream of the systemic memory of CD8+T cells.

## Enhancement of CD8+T$_{CM}$ cell differentiation after the combination therapy does not occur in TDLN but mainly in peripheral blood

Based on the activation of CD8+T cells after the combination therapy in the TDLN, we hypothesized that the differentiation of CD44$^{hi}$CD62L+CD8+T cells was enhanced in the TDLN compared with MWA + PBS before their migration into the peripheral blood. Thus, the TDLNs were obtained on day 35 for flow cytometry (Fig. 6a). Higher percentages and absolute numbers of tetramer+CD8+T cells were

**Fig. 5 | Immune microenvironment in TDLN is more activated by the combination therapy compared with local ablation alone. a** Experimental design for (**b**–**d**). Dissection of TDLN or sham surgery was performed on all mice on day 10. **b**–**d** Percentages of CD4+T, CD8+T, and CD44hiCD62L+CD8+T cells of mice (*n* = 4 per group). **e** Experimental design for (**f**–**i**). All mice were intraperitoneally injected with 2DG every 2 days from day 10 to day 30. FTY720 or 0.9% saline was intraperitoneally injected q.d from day 9 to day 35. **f** Percentages and absolute numbers of CD8+T, tetramer+CD8+T, CD44hiCD62L+tetramer+CD8+T cells in 4T1 model (*n* = 3 per group). **g** Percentages and absolute numbers of CD8+T, CD44hiCD62L+CD8+T cells in Py8119 model (*n* = 3 per group). **h, i** Representative flow plots of CD8+T and tetramer+CD8+T cells in (**e**). **j**–**l** WT BALB/c mice implanted with 4T1 were treated with MWA on day 10. All mice were intraperitoneally injected with 2DG or PBS every 2 days from day 10. TDLNs were collected on day 15 for RNA-seq (*n* = 5 per group). Estimated percentages of immune cells in TDLNs of mice (**j**). GO pathways enriched

in differentially expressed genes of TDLNs between the MWA + 2DG and MWA + PBS groups (**k**). Mean scores of GO pathways related to CD8+T and Th1 cells in the TDLNs (**l**). **m** Experimental design for (**n**–**q**). **n** Percentages of Th1, CD8+T, CD25+CD8+T, CD69+CD8+T, and CD44+CD8+T cells in TDLNs of mice (*n* = 4 per group). **o** Representative flow plots of migratory cDC1s in TDLNs of mice. **p, q** Percentages and absolute numbers of DCs, cDC1s, cDC2s, migratory cDC1s, and resident cDC1s in TDLNs of mice (*n* = 3 per group). The experiments (**a**–**i** and **m**–**q**) were repeated for at least two times. Data are mean ± SD (**b**–**d, f, g, n, p, q**), mean with min to max (**j**). Significance determined by two-tailed unpaired t-test (**b**–**d, f, g, j, n, p, q**) and the hypergeometric test (**k**). ns, not significant, *$P < 0.05$, **$P < 0.01$, ***$P < 0.001$, ****$P < 0.0001$. Exact *p* values and source data are provided as a Source Data file. **a, e, m** Created with BioRender.com released under a Creative Commons Attribution-NonCommercial-NoDerivs 4.0 International license.

observed in the MWA + 2DG group than those in the MWA + PBS group (Fig. 6b, c). Interestingly, no significant difference was found in either the percentages or absolute numbers of CD44hiCD62L+tetramer+CD8+T cell subset and CD44hiCD62L+CD8+T cell subset in the TDLNs of 4T1 and Py8119 models between the two groups (Fig. 6d, e), which meant 2DG might not exert its effect in the TDLN on the enhanced CD44hiCD62L+CD8+T cell differentiation after the combination therapy compared to MWA + PBS. Based on the fact that lymphocytes are re-circulating between different lymphoid organs and the peripheral blood, we speculated that the combination therapy might enhance CD44hiCD62L+CD8+T cell differentiation after their migration into the peripheral blood. We then isolated the PBMCs from mice 5 days after MWA (Fig. 6f). To explore the differentiation status of CD8+T cells after the combination therapy in the peripheral blood, 2DG was added or not to culture with the PBMCs in vitro. In line with our hypothesis, the percentage of CD44hiCD62L+CD8+T cells significantly increased in the 2DG group (Fig. 6g), further indicating that the peripheral CD8+T cells after MWA could be promoted to differentiate into CD44hiCD62L+CD8+T cells under the effect of 2DG. However, activated CD8+T cells in TDLN could also be promoted to differentiate into CD44hiCD62L+CD8+T cells by 2DG as peripheral CD8+T cells did (Fig. 6f, h). The activated CD8+T cells in our ex-vivo experiments of TDLNs should have migrated into the peripheral blood if they were still in vivo, which explained the discrepancy. To further determine whether the enhancement of differentiation took place in the TDLN, FTY720 was used to block T cells from exiting TDLN in the MWA + 2DG and MWA + PBS groups (Fig. 6i). Consistently, no significant differences in the percentages and absolute numbers of CD44hiCD62L+tetramer+CD8+T cells in the TDLNs of 4T1 models and CD44hiCD62+CD8+T cells in the TDLNs of Py8119 models were found between the 2 groups (Fig. 6j). To further determine the differentiation trajectory of tumor-specific CD8+T cells, CD8+T cells in the peripheral blood were collected from mice treated with MWA + PBS and MWA + 2DG on day 35 (Supplementary Fig. 6a). CD8+T cells were incubated with oligo-barcoded AH1-specific tetramer. Then, single-cell RNA seq and single-cell TCR seq were performed on these CD8+T cells. In shared TCR clonotypes of CD8+T cells between the two groups, tetramer-positive clonotypes of CD8+T cells were determined as tumor-specific CD8+T cells for further analysis (Fig. 6k). Consistently, a higher percentage of tumor-specific CD8+TCM was observed in the MWA + 2DG group compared to the MWA + PBS group (Fig. 6l). By pseudo-time analysis, the trajectory of tumor-specific CD8+T cell differentiation was investigated. The results indicated that glycolysis inhibition promoted more tumor-specific CD8+TCM differentiation from the early activated tumor-specific CD8+T cells (Fig. 6m and Supplementary Fig. 6c). Overall, a higher proportion of activated CD8+T cells with an equal ratio of CD44hiCD62L+CD8+T cell subset might be released into peripheral blood from TDLN, and CD8+TCM cell differentiation might be enhanced peripherally by 2DG compared with MWA alone.

## STAT1 signaling determines the memory differentiation of CD8+T cells with the remodeling of metabolism throughout the combination therapy

As 2DG was a modulator of cellular metabolism, it was necessary to explore the changes in the metabolism pathway of CD8+T cells in the MWA + 2DG group compared to the MWA + PBS group, which might help us a lot to understand the phenotype of CD8+T cells. Splenic CD8+T cells were isolated on day 35 from mice in the MWA + 2DG and MWA + PBS groups as described above (Fig. 6n, o, Supplementary Fig. 6a). RNA-seq was performed on the CD8+T cells for further analysis. Consistently, the memory scores of CD8+T cells were elevated in the MWA + 2DG group compared to the MWA + PBS group (Supplementary Fig. 6b). After the combination therapy, glycolysis, glycine, serine, and threonine metabolism, and the glutamine family amino acid catabolic process were significantly down-regulated compared to those in the MWA + PBS group (Fig. 6n). However, the changes in metabolism might be secondary to the different memory generation of CD8+T cells between the MWA + 2DG and MWA + PBS groups. Expression of metabolic genes was detected by quantitative polymerase chain reaction (qPCR) of CD8+T cells isolated from spleens of mice on day 15 in the MWA + PBS and MWA + 2DG groups and similar results were obtained (Supplementary Fig. 6d). Moreover, the results indicated that CD8+T cells over course of MWA + 2DG relied more on OXPHOS compared to those over course of MWA + PBS, which was consistent with our previous hypothesis (Supplementary Fig. 6d).

To further investigate what mechanism was at play in promoting the memory formation of CD8+T cells after the combination therapy, the transcription factors were estimated[41]. The results indicated that STAT1 might play an important role in the memory formation of CD8 + T cells induced by the combination therapy (Fig. 6o and Supplementary Fig. 6e). The splenic CD8+T cells were isolated 7 days after MWA to culture in vitro. Fludarabine, a STAT1 inhibitor, significantly inhibited the memory formation of CD8+T cells 4 days after coculturing with 2DG (Fig. 6p). To make our results more convincing, CD8-specific genetic interference of STAT1 was performed in vivo by CD8a-Cre mice and AAV-ark313[42] with the double-floxed inverted orientation (DIO) system (Fig. 6q, r). The shRNA target at Stat1 and Zsgreen only express in Cre+cells (CD8a+cells in our mouse model) but not in other cells. The percentages of CD44hiCD62L+subset decreased with the in-vivo knockdown of STAT1 in CD8+T cells from the MWA + 2DG group (Fig. 6s). Overall, the STAT1 signaling determined the memory differentiation of CD8+T cells with the remodeling of metabolism throughout combination therapy.

## Peripheral CD8+T cells of patients post MWA of local tumors differentiate into CD8+TCM cells by inhibiting glycolysis in vitro

To further validate the systemic memory phenotype of the combination therapy, three early-stage breast cancer patients from our clinical trial (cohort 1, ChiCTR2000029155) who received MWA were enrolled

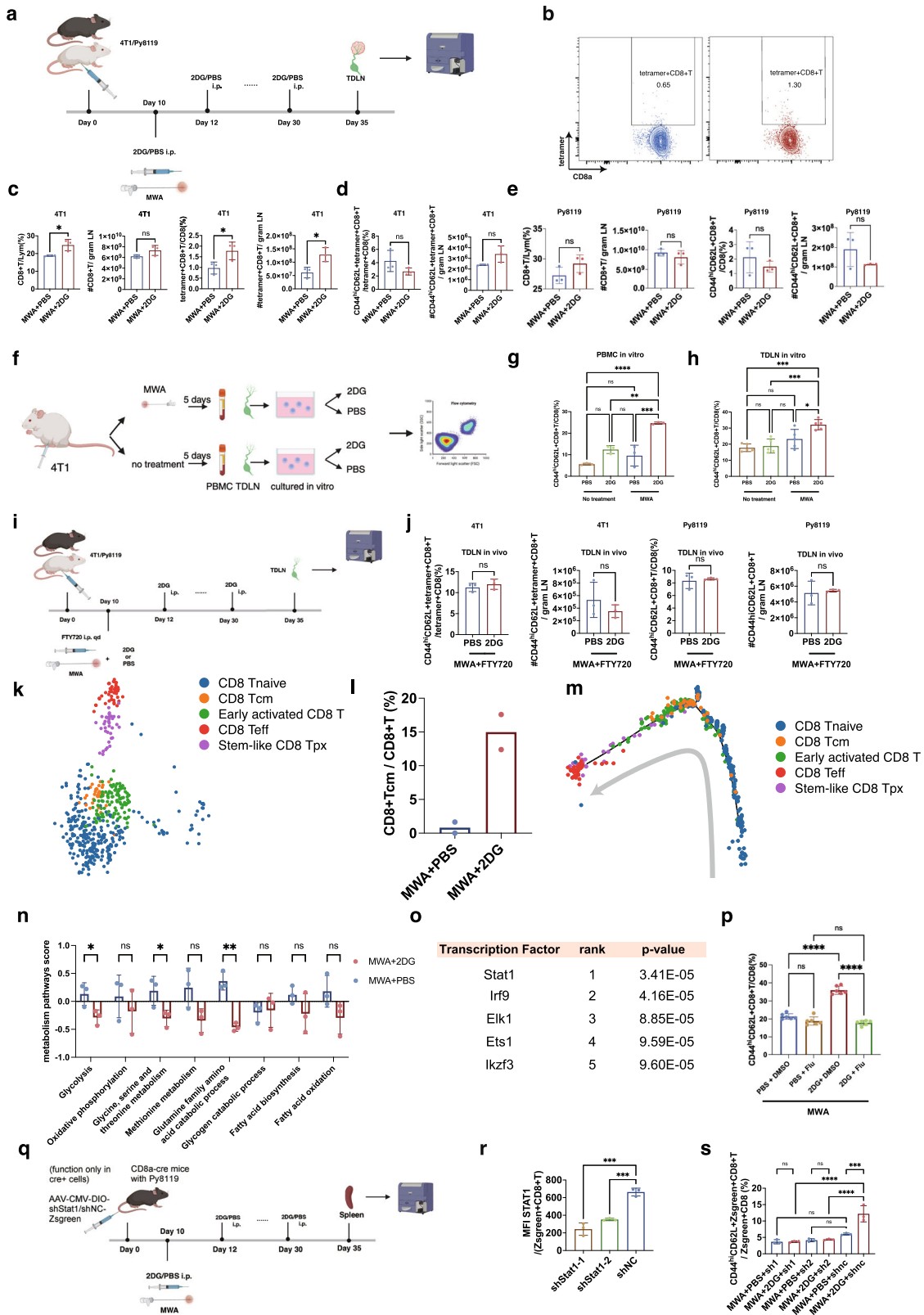

(Fig. 7a). Given that the enhancement of CD8⁺T_CM cell differentiation mainly took place in the peripheral blood after the combination therapy in the mouse model, we collected the peripheral blood from the patients 1 day before and 7 days after MWA to further validate our results (Fig. 7a). PBMCs were then isolated to culture in vitro with or without 2DG for 4 days to simulate the combination therapy (Fig. 7a). The percentages of CD8⁺T cell subsets were analyzed by flow

cytometry (Fig. 7a, Supplementary Fig. 7a). CD39 was utilized to detect tumor-specific T-cell clones in our ex-vivo experiments, which was capable of discriminating tumor-reactive T-cell clones from bystander clones such as virus-specific T cells from the tumor or periphery[43,44]. Compared to the PBMCs cultured without 2DG before MWA, the PBMCs before ablation cultured with 2DG showed increased percentages of CD8⁺T_CM and CD39⁺CD8⁺T_CM cells in vitro. Moreover, the

**Fig. 6 | Enhancement of CD8⁺T$_{CM}$ cell differentiation after the combination therapy does not occur in TDLN but mainly in peripheral blood. a** Experimental design for (**b–e**). **b** Representative flow plots of tetramer⁺CD8⁺T cells in TDLNs of mice in (**a**). **c, d** Percentages and absolute numbers of CD8⁺T, tetramer⁺CD8⁺T, CD44^{hi}CD62L⁺tetramer⁺CD8⁺T cells ($n = 3$ per group) implanted with 4T1. **e** Percentages and absolute numbers of CD8⁺T, CD44^{hi}CD62L⁺CD8⁺T cells ($n = 3$ per group) implanted with Py8119. **f** Experimental design for (**g, h**). PBMCs (**g**) or single cells of TDLNs (**h**) were collected for in-vitro experiments. Percentages of CD44^{hi}CD62L⁺CD8⁺T cells in PBMCs (**g**, $n = 3$) or single cells of TDLNs (**h**, $n = 5$) cultured in vitro. **i** Experimental design for (**j**). **j** Percentages and absolute numbers of CD44^{hi}CD62L⁺tetramer⁺CD8⁺T and CD44^{hi}CD62L⁺CD8⁺T cells ($n = 3$ per group). **k** Uniform manifold approximation and projection (UMAP) plot of tumor-specific CD8⁺T cells from the 2 groups. **l** Percentages of CD8⁺T$_{CM}$ in the single-cell RNA seq data. **m** Monocle2 pseudotime trajectory showing the differentiation of tumor-specific CD8⁺T cells. **n, o** Metabolism pathway scores of CD8⁺T cells ($n = 3$ per group) and estimated transcription factor promoting memory formation of CD8 + T cells ($n = 3$ per group). **p** Percentages of CD44^{hi}CD62L⁺CD8⁺T cells in isolated CD8⁺T cells after coculturing with PBS + DMSO, PBS + fludarabine (flu), 2DG + DMSO, or 2DG + fludarabine for 4 days ($n = 6$ per group). **q** Experimental design for (**r–s**). **r** Knockdown of total STAT1 of Zsgreen⁺CD8⁺T cells in spleens of CD8a-cre mice injected with AAV ($n = 3$ per group). **s** Percentage of CD44^{hi}CD62L⁺Zsgreen⁺CD8⁺T cells ($n = 3$ per group). The experiments (**a–j** and **p–s**) were repeated for at least two times. Data are mean ± SD (**c–e, g, h, j, l, n, p, r, s**). Significance determined by two-tailed unpaired t-test (**c–e, j, n**), one-way ANOVA (**g, h, p, r, s**), and one-sided Wilcoxon rank-sum test (**o**). ns, not significant,*$P < 0.05$, **$P < 0.01$, ***$P < 0.001$, ****$P < 0.0001$. Exact p values and source data are provided as a Source Data file. **a, f, i, q** Created with BioRender.com released under a Creative Commons Attribution-NonCommercial-NoDerivs 4.0 International license.

PBMCs after MWA cultured with 2DG showed significantly higher percentages of CD8⁺T$_{CM}$ and CD39⁺CD8⁺T$_{CM}$ cells than the other three groups (Fig. 7b, c, Supplementary Fig. 7b). To further validate our results, another nine consecutive patients with breast cancer from our phase II clinical trial (cohort 2, NCT04805736) who only received MWA were enrolled. Similar results were obtained (Fig. 7b, d, Supplementary Fig. 7b). In clinical practice, local ablation is used as a standard therapy for liver tumors. Ex-vivo experiments were also replicated in the liver tumor (Fig. 7e, cohort 3). Peripheral blood before and 3–9 days after MWA was collected to isolate PBMCs. PBMCs were treated with 2DG or not as described above (Fig. 7e). The PBMCs before ablation cultured with 2DG showed significantly higher percentages of CD8⁺T$_{CM}$ and CD39⁺CD8⁺T$_{CM}$ cells compared to the PBMCs cultured without 2DG before MWA (Fig. 7f, g, Supplementary Fig. 7b). Consistently, the PBMCs cultured with 2DG after ablation showed significantly higher percentages of CD8⁺T$_{CM}$ and CD39⁺CD8⁺T$_{CM}$ cells compared to the other three groups (Fig. 7f, g, Supplementary Fig 7b). The results indicated that inhibiting glycolysis after MWA of human malignant tumors enhanced the memory phenotype of peripheral CD8⁺T cells.

## Discussion

Minimally invasive thermal therapy has been gradually applied to treat solid tumors as standard local therapy, including hepatocellular carcinoma, lung cancer, renal carcinoma, thyroid tumor, and so on[4–8]. However, recurrence may still occur after local ablation of tumors. To our knowledge, many studies have reported the immune response induced by tumor ablation, and nearly all of them have concluded that the immune response is weak. In our study, the combination therapy could establish a relatively long-term systemic anti-tumor effect, which completely abolished rechallenged tumor growth in both immunological "hot" MC38 and "cold" B16F10, 4T1, Py8119 tumor models. Notably, we found that MWA alone still exhibited a very strong anti-tumor effect in the immunological "hot" tumor models but not in the immunological "cold" tumor models. This finding is meaningful because it further proves the essentiality of applying the combination therapy to immunological "cold" tumors. To the best of our knowledge, many studies have proposed different combinational strategies to enhance the efficacy of tumor ablation only by verifications in mouse models[45–47]. However, clinical validation was performed on PBMCs obtained from patients with immunologically different breast or liver tumors by the ex-vivo experiments in our study. Different from previously reported partial ablation research[35–37], our study focused on complete local ablation, which is widely used on early-stage malignant tumors in clinical practice. Our data also revealed that postablative glycolysis inhibition promoted more differentiation of CD8⁺T$_{CM}$ cells, which might be attributed to the activation of CD8⁺T cells induced by ablation. The result indicated the significance of tumor ablation in the combination therapy. Thus, we proposed the strategy of combining

glycolysis inhibition with local ablation, by which the immunogenicities of cold tumors might be improved for better adaptive immune response.

2DG, a glycolysis inhibitor used in the study, has proven safe and non-toxic in several preclinical studies and clinical trials to treat solid tumors alone or combined with other anti-cancer treatments, even with repeated administration plans[38,48–52]. The most common adverse events after 2DG were not life-threatening including fatigue, sweating, dizziness, and nausea, mimicking the symptoms of hypoglycemia, and grade 3 or 4 adverse events related to 2DG occurred in few patients[38,51]. Based on the Warburg effect of tumor, radiolabeled 2DG (2-deoxy-2-[18 F]fluoro-d-glucose) is used in Positron Emission Computed Tomography due to its preferential accumulation in the tumor. 2DG also exploits viral-induced increased glucose uptake and metabolism by blocking glycolysis and interfering with N-linked glycosylation, promoting emergency approval of 2DG in India to be used against SARS-CoV-2[53]. As the metabolic switch of T cells toward aerobic glycolysis upon their activation by TCR-signaling, 2DG can also accumulate in activated T cells during cotreatment regimens. Glycolysis inhibition can reduce the proliferation of stimulated CD8⁺T cells. Thus, mice were injected with 2DG at a long interval of 2 days in our study to prevent the immunosuppressive effect of glycolysis inhibition. Subsequently, our data showed that the combination therapy didn't reduce the overall anti-tumor effect compared to the MWA alone. Moreover, in our data, a more robust memory phenotype of CD8⁺T cells was induced by the combination therapy. As the accumulation of 2DG in tumor cells, the growth of the tumor can be delayed by blocking glycolysis. Although we excluded 2DG's influence on tumors in our study, we might speculate that the combination therapy was beneficial to the therapeutic outcome clinically by both 2DG's direct effect on tumor cells and the robust immune responses induced by the combination therapy. Overall, it is plausible that 2DG could be considered as the potential adjuvant agent to inhibit glycolysis after ablation, which remains to be explored clinically in the future.

Memory T cells can respond to 100-fold lower doses of antigen and respond more rapidly in the presence of costimulation within 0.5 h to 2 h compared with naïve T cells[54,55]. Mechanically, we demonstrated that the combination therapy enhanced the memory phenotype by enhancement of CD8⁺T$_{CM}$ cell differentiation. To our surprise, the memory phenotype was even further enhanced as time progressed with more CD8⁺T$_{CM}$ cells after the combination therapy. In some immunotherapies such as the CAR-T therapy[56,57], T cells with memory signatures from patients are identified as one paramount determinant of therapeutic success, which indicates the promising therapeutic effect of the combination therapy. However, the combination therapy may be meaningless if the enhanced anti-tumor immunity is nonspecific. The naturally occurring tumor-specific T cells are the most suitable immune cells for anti-cancer therapy because of their high response rate and high safety[58,59]. In our study, the combination

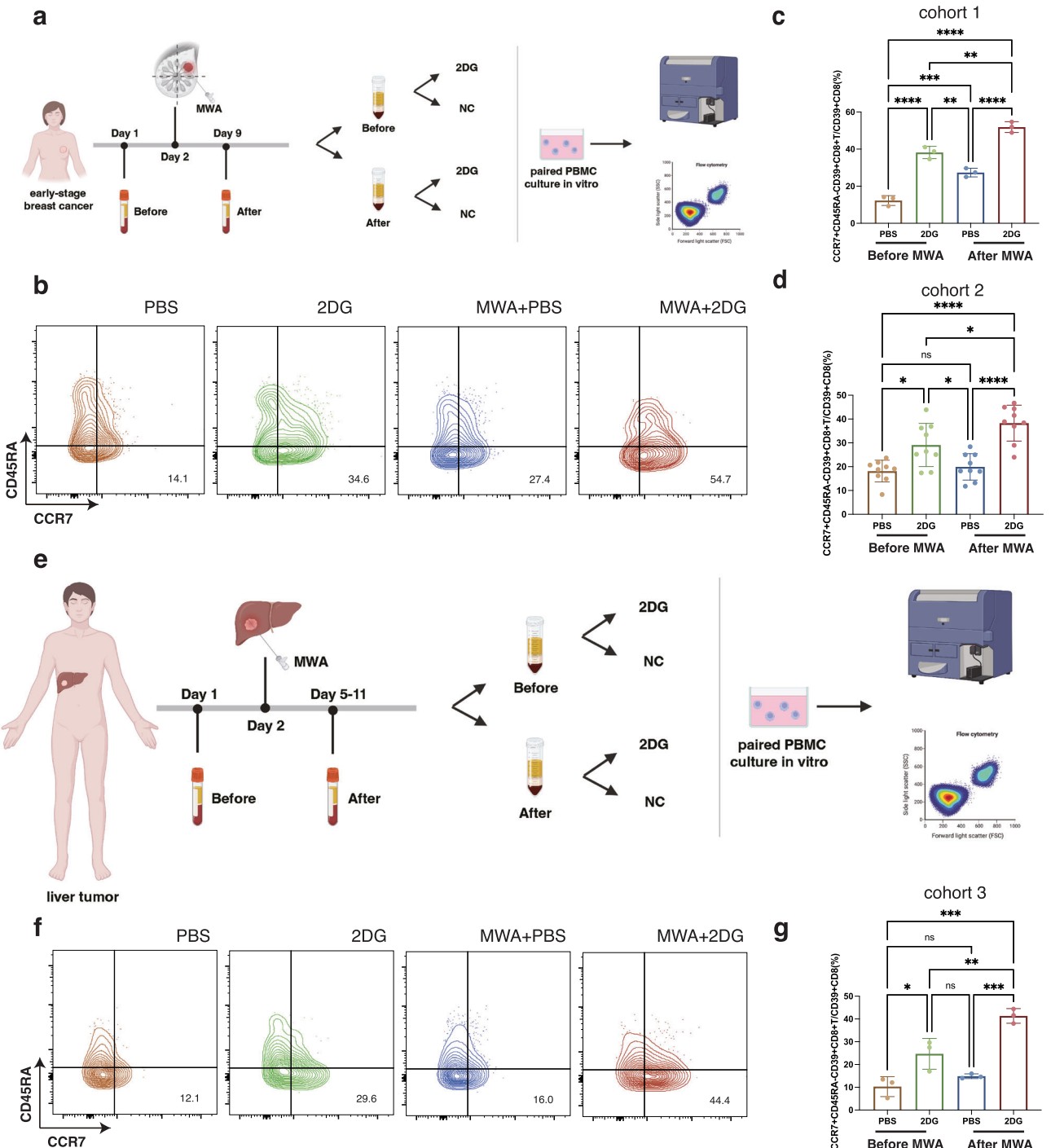

**Fig. 7 | Peripheral CD8 + T cells of patients post MWA of local tumors differentiate into CD8 + T_{CM} cells by inhibiting glycolysis in vitro. a** Experimental design for (**b**–**d**). Peripheral blood was collected 1 day before and 7 days after MWA of breast cancer. PBMCs isolated from the peripheral blood were cultured in vitro with 2DG or vehicle in a paired way. **b** Representative flow plots of CD39$^+$CD8$^+$T_{CM} cells in the 4 groups of PBMCs in (**a**) cultured in vitro. **c, d** Percentages of CD39$^+$CD8$^+$T_{CM} cells in the 4 groups of PBMCs in (**a**) cultured in vitro (n = 3 per group in cohort 1 and n = 9 per group in cohort 2). **e** Experimental design for (**f, g**). Peripheral blood was collected 1 day before and 3–9 days after MWA of liver tumor.

PBMCs isolated from the peripheral blood were cultured in vitro with 2DG or vehicle in a paired way. **f** Representative flow plots of CD8$^+$T_{CM} cells in the 4 groups of PBMCs in (**e**) cultured in vitro. **g** Percentages of CD8$^+$T_{CM} cells in the 4 groups of PBMCs in (**e**) cultured in vitro (n = 3 per group). Data are mean ± SD (**c, d, g**). Significance determined by one-way ANOVA (**c, d, g**). ns, not significant,*P < 0.05, **P < 0.01, ***P < 0.001, ****P < 0.0001. Exact p values and source data are provided as a Source Data file. **a, e** Created with BioRender.com released under a Creative Commons Attribution-NonCommercial-NoDerivs 4.0 International license.

therapy significantly increased the percentage of tumor-specific CD44$^{hi}$CD62L$^+$CD8$^+$T cells compared with MWA alone, which meant a more potent anti-tumor effect was induced. As more tumor-specific CD8$^+$T cells were observed in the MWA + 2DG compared to the other groups, tumor-specific immunity might be enhanced by glycolysis

inhibition. However, the mechanisms are still unclear, which will be further explored in the near future.

TDLN is pivotal in generating effective anti-tumor T-cell responses because of the draining or transport of tumor antigens to TDLNs for further T-cell activation. Others have previously hinted toward a role

for TDLN in the generation of systemic anti-tumor immunity induced by different therapies in the murine models[60–63]. Our data showed that the deficiency of TDLNs resulted in the attenuated systemic memory phenotype of CD8⁺T cells and reduced tumor-specific CD8⁺T cells induced by the combination therapy, which is consistent with the theory that tumor-reactive lymphocytes exit from the TDLNs.

Several clinical trials of both melanoma and breast cancer indicated that lymph node dissection did not provide an overall survival benefit with sentinel lymph node involvement[64–66]. Moreover, our previous study[67] has found that patients with stage IV distant lymph node metastases show similar survival to those with stage III breast cancer, suggesting lymph node involvement may be not life-threatening. It is conceivable that the dissection of TDLNs should be carefully considered under circumstances where resection may actually be deleterious for the abscopal effect induced by systemic therapy such as immunotherapy. Thus, the conservation of TDLNs might become a new trend in the future because of their important role in the generation of systemic anti-tumor immunity.

Several limitations still existed. First, due to the clinical validation being performed in vitro, future clinical trials are needed to prove the long-term anti-tumor effect induced by combination therapy in distinct types of tumors. Second, it remains unclear whether other glycolysis inhibitors can enhance the systemic anti-tumor effect induced by MWA. More investigations should also be performed on the sequence of MWA and glycolysis inhibition in the future. Third, the increased ratio of tumor-specific CD8⁺T cells might be attributed to the tumor antigen released after MWA. However, due to the lack of antibodies against tumor antigens, we didn't examine the release of tumor antigens after MWA. 2DG also played an important role in tumor-specific immune responses, and the underlying cause should be investigated in the future. Fourth, the combination therapy established a robust anti-tumor effect on the basis of no metastasis within TDLN. The therapeutic effect of the combination therapy deserves further investigation under circumstances where TDLNs are involved. Moreover, the essentiality and therapeutic outcomes of the dissection of TDLNs should be considered carefully in the future.

In conclusion, the memory phenotype of CD8⁺T cells after MWA of human breast cancer increased slightly, but the metabolism status of CD8⁺T cells after MWA was unfavorable for memory generation. Our work offered a potential strategy of the combination of tumor ablation and glycolysis inhibition in several tumor models, which exhibited promising anti-tumor effects via CD8⁺T_{CM} cells. Mechanically, enhancement of CD8⁺T_{CM} cell differentiation was TDLN dependent but took place in the peripheral blood determined by STAT1. Importantly, the peripheral CD8⁺T cells from patients with breast or liver tumors after MWA were promoted to differentiate into CD8⁺T_{CM} cells by inhibiting glycolysis in vitro. In summary, glycolysis inhibition after tumor ablation may be a promising treatment strategy for early-stage malignant tumors, although this strategy should be tested in future clinical trials.

## Methods

### Study cohorts

A clinical trial (ChiCTR2000029155) performed to determine the short-term local effect of MWA in the treatment of early-stage breast cancer has been previously described[17]. Another clinical trial (NCT04805736) was performed to determine the feasibility of MWA combined with anti-PD1 therapy in patients with early-stage breast cancer. Three consecutive patients with liver tumors treated with MWA were enrolled at our hospital. In our study, PBMCs were collected from patients in the above 3 cohorts treated with MWA for only in-vitro experiments. All patients enrolled were provided with written informed consent. All obtained samples were approved by the institutional ethics committee of The First Affiliated Hospital of Nanjing Medical University. All patients with breast tumors were female in our

study, while one patient with liver tumors was male and the other two were female. Sex was determined by investigator observation. Only one male patient was included thus sex gender analysis was not carried out.

### Mice

Six weeks or 7 months (control group for the mice lived for 140 days after MWA + 2DG) WT female BALB/c, 6 weeks WT female C57BL/6, 6 weeks female BALB/cAnSlac-nu, and 4–8 weeks female Tg(Cd8a-iCre) mice were maintained under specific pathogen-free conditions in the Animal Core Facility of Nanjing Medical University and housed under a 12-h light/dark cycle and given ad libitum access to food and water. All mice used in our study were female. BALB/c (strain NO. 211) and C57BL/6 (strain NO. 213) mice were purchased from Charles River. BALB/cAnSlac-nu mice (stock NO. LSF-05) were purchased from the Animal Core Facility of Nanjing Medical University. Tg(Cd8a-iCre) mice (strain NO. T007005) were generated by GemPharmatech (Nanjing, China). Animals were randomly assigned to experimental groups. All mice were used following the protocols (IACUC-2202022) reviewed and approved by the Institutional Animal Care and Use Committee (IACUC), Animal Core Facility of Nanjing Medical University.

### Cell lines

The BALB/c mice-derived mammary carcinoma 4T1 cell line and C57BL/6 mice-derived mammary carcinoma Py8119 cell line were purchased from the American Type Culture Collection (ATCC, USA). The C57BL/6 mice-derived melanoma B16F10 cell line was purchased from Meisen cell technology Co. Ltd., Meisen Chinese Type Culture Collection (MeisenCTCC, Hangzhou, China). The C57BL/6 mice-derived colon adenocarcinoma cell line MC38 was purchased from Kerafast (USA). 4T1 and B16F10 cell lines were cultured in Dulbecco's modified Eagle's medium (DMEM, Hyclone, USA) supplemented with 100 units/ml penicillin, 100 μg/ml streptomycin (Hyclone, USA) and 10% (v/v) fetal bovine serum (FBS, Hyclone, USA). MC38 cell line was cultured in RPMI-1640 (Hyclone, USA) supplemented with the above contents. Py8119 cell line was cultured in F-12K (Hyclone, USA) supplemented with the above contents. All cells were cultured in a 37 °C incubator with 5% CO₂. For generation of 4T1-zsgreen, plasmid (PGMLV-CMV-MCS-EF1-ZsGreen1-T2A-Neo) was used to package the lentivirus. 4T1 cells were then transfected with lentivirus with 8 μg/mL polybrene (Genepharm) and then selected after G418 (Beyotime) treatment for 2 weeks. Mycoplasma was confirmed negative by the standard PCR method.

### Tumor models

4T1 or Py8119 tumor cells were injected subcutaneously into the right fourth mammary fat pad at a density of $5 \times 10^5$ cells in 100 μL PBS on day 0. B16F10 and MC38 murine cells were injected subcutaneously into the right inguinal flank at a density of $5 \times 10^5$ cells in 100 μL PBS on day 0. To simulate early-stage tumors, no apparent metastases were found on day 10 after tumor cell inoculation in multiple tumor models according to preliminary data. To simulate metastasis in distant places, 5 or $1 \times 10^5$ 4T1 or Py8119 tumor cells were injected subcutaneously into the left fourth mammary fat pad on day 35 or day 140. $5 \times 10^5$ B16F10 or MC38 were injected subcutaneously at the contralateral inguinal flank on day 35. Tumor volume (mm³) was calculated by the following formula: $V = \pi/6 \times L \times H^2$, where $H$ is the shorter diameter and $L$ is the longer diameter. Tumor size was measured by calipers every five days. The mice were immediately euthanized by cervical dislocation when the tumor size reached about 1.5 cm in diameter according to animal ethics.

### Mouse treatments

Mice were implanted with 4T1, Py8119, MC38, and B16F10 tumor cells inoculations on day 0. On day 10, mice were randomly divided into

four groups: PBS treated group (PBS), 2DG treated group (2DG), MWA + PBS treated group (MWA + PBS) and MWA + 2DG treated group (MWA + 2DG). In the 2DG-treated group, 500 mg/kg 2DG (HY-13966, MedChemExpress, China) dissolved in PBS was injected intraperitoneally into each mouse every 2 days from day 10 to day 30. Mice injected with the same volume of PBS were set as control. For MWA, an output power of 5 W was applied, using a microwave generator (ECO-100E, Yigao Microwave Electric Institute, Nanjing, China) with an irradiation frequency of 2450 MHz, as previously reported[68]. In the MWA + 2DG or MWA + PBS group, mice were injected with 2DG or PBS as above immediately after MWA of primary tumors. For the CD8+T cell depletion experiment, each mouse after the combination therapy was injected intraperitoneally with 200 µg anti-CD8a (clone: 53-6.7, Biolegend, USA) or control rat antibody IgG2a, κ (Biolegend, USA) for five times every four days. The depletion was started one day before tumor cells re-challenge and continued during the experiment. For mice with TDLN resection, the inguinal lymph node at the same side of the tumor was resected with a stereomicroscope. The location of the incision was about 0.5 cm above the hip joint of the mice. The wound was sutured with 4/0 nylon. For FTY720 (HY-12005, MedChemExpress, China) injection, 3 mg/kg FTY720 dissolved in the 0.9% saline was injected intraperitoneally into each mouse every day. Mice injected with the same volume of 0.9% saline were set as control. For the adoptive transfer of non-CD8 immune cells, mice were randomly divided into four groups: MWA + 2DG + 2DG treated non-CD8 immune cells, MWA + 2DG + PBS treated non-CD8 immune cells, MWA + PBS + 2DG treated non-CD8 immune cells, MWA + PBS + PBS treated non-CD8 immune cells. Immune cells of donor mice treated with MWA + 2DG or MWA + PBS described above were collected 5 h after the injection of 2DG or PBS every 2 days from day 10 to day 30. The isolated immune cells with CD8+T cells depleted were then transferred into the recipient mice treated with MWA + 2DG or MWA + PBS described above. The mice received $5 \times 10^6$ cells intravenously every 2 days from day 10 to day 30. For CD8a-specific genetic interference, CD8a-cre mice were injected intravenously with 200 µL HBAAV2/ARK313-CMV-DIO-Mir30-1-m-STAT1-ZsGreen (1.3*10^12vg/mL), HBAAV2/ARK313-CMV-DIO-Mir30-2-m-STAT1-ZsGreen (1.2*10^12vg/mL) or HBAAV2/ARK313-CMV-DIO-ZsGreen (1.0*10^12vg/mL). The AAVs were generated by Hanbio Biotechnology Co., Ltd (Shanghai, China). The shRNA sequences used are listed in the supplementary table. 3. All procedures in vivo were aseptic.

### Detection of phagocytosed zsgreen and released zsgreen
For in-vitro experiments, thermal treatment was performed on 4T1-zsgreen cells cultured in 10 cm dishes with a cell density of 80–90%. The culture medium of 4T1-zsgreen was replaced by 3 ml PBS in different thermal treatment groups. The dishes were sealed with parafilm and submerged in a water bath at different temperatures for 3 min. PBS without tumor cells was collected carefully for centrifugation, and the supernatant was collected for fluorescence analysis (TECAN SPARK).

For in-vivo experiments, tumor tissues and TDLNs with the same volume from the MWA or non-treatment groups were collected. Tissues were carefully cut into small pieces avoiding cell damage with the minimum manipulation and were eluted in PBS with the same volume. The PBS without tissues was collected carefully for centrifugation, and the supernatant was collected for fluorescence analysis (TECAN SPARK). Released zsgreen captured by DCs or macrophages in TDLNs were evaluated by flow cytometry.

### UPLC-ESI-MS/MS
2DG was configured with ammonia water into the following concentrations: 50, 100, 200, 500, 1000, 2000, 5000, and 10,000 ng/mL working solution. 0.2 mol 1-phenyl-3-methyl-5-pyrazolone (PMP, 200 µL) was added to 200 µL working solution for further

derivatization at 70 °C for 30 min, which was regarded as standard samples. 100 µL mice plasma was added with 400 µL 80% acetonitrile. After centrifugation with 8000 g for 10 min, the supernatant was concentrated and dried under vacuum and was resolubilized with 200 µL ammonia water. Samples were then derivatized with PMP as standard samples did. Samples were analyzed by an Xevo TQ-S micro triple quadrupole mass spectrometer (Waters). With a flow rate of 0.3 ml/min at 35 °C, BEH C18 analytical column (2.1 mm × 50 mm, 1.8 µm) was used to perform chromatographic separation.

The following agents were used as the mobile phase: Formic acid (FA) in water (0.1%, v/v, 10 mM, solvent A) and acetonitrile (solvent B). A gradient of 0.5 min 5% B, 2 min 5–95% B, 1 min 95% B, 0.1 min 95–5% B, and 2.4 min 55% B was used. The following optimized mass parameters were set: desolvation gas flow, 800 L/h; interface voltage, 3 kV; collision-induced dissociation argon gas pressure; desolvation temperature, 500 °C; and heat block temperature, 150 °C; The mass transition for 2-DG-PMP set as m/z 495.29 > 321.14 (+). Masslynx was used for the data acquisition and analysis. The analysis was performed at Nanjing Positive Function Biotechnology Co., Ltd.

### Metastases analysis
Mice were injected subcutaneously with the 4T1 tumor cells as above. Mice were sacrificed 35 days after the inoculation, and the TDLNs and lungs were harvested. The number of tumors on the surfaces of lungs was counted to measure metastatic tumor burden. Both TDLNs and lungs were fixed in 4% formalin, embedded in paraffin, sectioned, and stained with Hematoxylin and eosin (H&E) to measure metastases further.

### Transmission electron microscopy
Tumor tissues after MWA were cut into small sizes of 1 mm³ for fixation with 2% glutaraldehyde and 1% OsO4. The samples were then dehydrated, embedded, cut into ultrathin sections, stained with 2% uranyl acetate, and stained with lead citrate. Images were acquired with a transmission electron microscope (Hitachi, HT7800/HT7700).

### Preparation of the single-cell suspension of spleen and TDLN
After mice were sacrificed, the spleens and the TDLNs were collected to isolate single cells for later use.

For single splenocytes, the harvest spleens were crushed gently with the plunger of a 10-mL syringe and passed through a 70-µm nylon mesh cell strainer (Biosharp, China). Then the cells were treated with an erythrocyte-lysing reagent (MULTISCIENCES, China) to remove red blood cells.

For single cells of TDLN, the harvested TDLN was also crushed gently with a 70-µm nylon mesh cell strainer (Biosharp, China). Single cells of TDLN and single splenocytes were resuspended with PBS for further analysis.

### PBMCs isolation
For patients, peripheral blood was collected 1 day before MWA and 3–9 days after MWA. EDTA-K2 tubes were used to collect peripheral blood. Then, PBMCs were isolated by Ficoll discontinuous density gradient centrifugation within 2 h of blood sample collection. The PBMCs were resuspended in PBS for later use.

For mice, EDTA-K2 tubes were used to collect blood retro-orbitally. PBMCs were isolated by Ficoll discontinuous density gradient centrifugation immediately. The PBMCs were resuspended in PBS for later use.

### CD8+T cell isolation and depletion
Single cells of spleens or PBMCs of mice were resuspended in the sorting buffer (Biolegend, Cat#480017) and were sorted by the mouse CD8+T cell isolation kit (Biolegend, Cat#480008) in accordance with the manufacturer's instructions.

Single cells of spleens of mice were resuspended in the sorting buffer (Biolegend, Cat# 480017) and were depleted by the mouse CD8a selection kit (Biolegend, Cat# 480136) in accordance with the manufacturer's instructions.

### Cultures of PBMCs, single cells from the TDLN, and CD8$^+$T cells

PBMCs, single cells of TDLNs, and CD8$^+$T cells were resuspended at $2 \times 10^6$ cells per mL in RPMI-1640 medium (Hyclone, USA) supplemented with 100 units/mL penicillin, 100 μg/mL streptomycin and 5% FBS.

For mice, the 96-well plates were precoated with 5 μg/ml anti-CD3ε (clone: 145-2C11, Biolegend) at 4 °C overnight or at 37 °C for 2 h. The RPMI-1640 medium was then added with 5 μg/ml soluble anti-CD28 (clone: 37.51, Biolegend) and 40 ng/ml recombinant mouse interleukin 2 (Cat#575402, Biolegend). Cells ($5 \times 10^5$) were incubated with different agents: 5-μM fludarabine (MedChemExpress, China), 2-mM 2DG (MedChemExpress, China) or vehicle in 96-well plates for 4 days. For humans, the 96-well plates were precoated with 2 μg/ml anti-CD3 (clone: HIT3a, Biolegend) at 4 °C overnight or at 37 °C for 2 h. The RPMI-1640 medium was then added with 1 μg/ml soluble anti-CD28 (clone: CD28.2, Biolegend) and 40 ng/ml recombinant human interleukin 2 (Cat# 589102, Biolegend). Cells ($5 \times 10^5$) were incubated with 2-mM 2DG (MedChemExpress, China) or vehicle in 96-well plates for 4 days. Cells were then harvested and washed twice in sterile PBS for later analysis.

### In vitro CTL cytotoxicity assay

4T1 cell line was seeded at a density of 5000 per well in 96-well plates 24 h prior to coculture. The isolated CD8$^+$ T cells 7 days after MWA and the tumor cells were co-cultured in RPMI medium supplemented with 5% FBS, 0.01% penicillin/streptomycin, 1 mM L-glutamine and 40 ng/mL IL-2 (Cat#575402, Biolegend). Anti-CD3 (clone: 145-2C11, Biolegend) and anti-CD28 (clone: 37.51, Biolegend) were used to stimulate CD8$^+$T cells. After 24 h coculture, the cytotoxicity of CTLs was assessed using Cytotoxicity LDH Assay Kit (dojindo, Cat# CK12).

### RNA extraction and RT-qPCR

Total RNA of CD8$^+$T cells was extracted using Trizol reagent (TaKaRa, Japan). The reverse transcribed using HiScript Q RT SuperMix (Vazyme, China), and RT−qPCR was performed on a real-time PCR instrument (Applied Biosystems, USA). The primers used are listed in the supplementary Table. 2.

### Oligo-barcoded tetramer preparation and incubation

Oligo-barcoded Streptavidin reagents (Biolegend, Cat# 405261) and Biotinylated H-2Ld MuLV gp70 Monomer-SPSYVYHQF (MBL, Cat# TB-M521-M) were used for generation of tetramers according to the Biolegend's instructions. CD8$^+$T cells were incubated with oligo-barcoded tetramers on ice in the dark for 30 min following Biolegend's instructions. The CD8$^+$T cells were harvested for single-cell RNA seq.

### Library preparation for scRNA-seq, scTCR-seq, and CITE-seq

For single-cell RNA seq of CD8$^+$T cells isolated from PBMCs of mice. The single-cell suspension was loaded onto a 10X Chromium A Chip and the scRNA-Seq libraries were constructed following the protocol of the 10X genomics Chromium Single Cell Immune Profiling Solution. Briefly, -20,000 cells (90−95% viability) were encapsulated into droplets at a concentration of 900 cells/μL. After the RT reaction, droplets were broken and barcoded-cDNA was purified with DynaBeads, followed by PCR-amplification. Sequencing libraries of single-cell transcriptomes suitable for Illumina sequencing platform were constructed after partial cDNA fragments and splicing. The remaining cDNA was enriched for the immune receptor (TCR), Cell Surface Protein libraries. Besides, the enriched products were amplified by PCR to construct a sequencing library suitable for the Illumina sequencing

platform. Finally, each library was sequenced on Illumina NovaSeq 6000 with 150 bp paired-end reads. Average sequencing depth aimed for the mRNA library is 30,000 read pairs per cell, 3000 read pairs per cell for the TCR libraries, and 5000 read pairs per cell for Cell Surface Protein libraries.

### Single-cell data analysis and processing

Data from our clinical trial (ChiCTR2000029155) was used in this study[17]. As previously reported, we processed the raw data with quality control and performed dimensionality reduction, clustering, cell-type labeling, and visualization. The memory phenotype of CD8$^+$T cells after MWA was further analyzed by IL7R. Two memory CD8$^+$T cell signatures were also applied (Fig 1f[32] and Supplementary Fig. 1c[33]). The metabolic signature gene sets were obtained from the KEGG (Kyoto Encyclopedia of Genes and Genomes) dataset (https://www.kegg.jp/) or the MsigDB database (http://www.gsea-msigdb.org/). The metabolic scores were calculated by R package GSVA (v 1.40.1). To further identify the metabolic differences before and after MWA in two CD8$^+$T clusters in the single-cell RNA data, R package limma (v3.48.3) was used. Delta score (Δscore) was used to further clarify the metabolism status of CD8$^+$T cells, defined as the D-value between glycolysis and OXPHOS.

For data of CD8$^+$T cells isolated from PBMCs of mice, the primary row data were converted to fastq format using the Illumina bcl2fastq converter and filtered to obtain clean data. The criteria included the following: 1, removal of polyA reads, 2, removal of reads containing more than 3 indeterminate bases, and 3, removal of low-quality reads (the number of bases with a Q value less than or equal to 5 that accounted for more than 20% of the total reads). Then, the clean data were processed using Cell Ranger software (version 7.0.0) provided by 10X Genomics to demultiplex cellular barcodes and align valid barcodes, and STAR was used to align the reads with the reference genome (mm10-2020-A). The gene expression pattern was measured by determining unique molecular identifier (UMI) counts using CellRanger.

Quality control and statistical analysis of single-cell transcriptome data using the Seurat package (v3.1.2) for R software (v3.5.1). The 1% of cells with the highest total number of genes and UMIs were removed, as well as cells with >10% of mitochondrial genes, and the filtered 56,671 cells were used for subsequent analysis in this project. Then single-cell expression data were normalized using the LogNormalize package, after which the FindVariableFeatures package was used to perform highly variable gene analysis using default parameters, and the highly variable genes after the above processing were subjected to principal component analysis. Cell clustering information was obtained by cell clustering analysis using (dims.use = 1:20, resolution = 1.2) through the FindClusters software package, which in turn resulted in the presentation of single-cell data using the UMAP (uniform manifold approximation and projection) algorithm. For the clustering of CD8$^+$T cells, the top 30 PCs were selected with a resolution parameter equal to 1.2. The cell type identification of each cluster was determined according to the expression of canonical markers from the reference database SynEcoSysTM (Singleron Biotechnology).

### TCR sequences assembly and analysis

TCR clonotype assignment were performed using cellranger(version 7.1.0) with (GRCm38_alts_ensembl) as reference. In brief, a TCR diversity metric, containing clonotype frequency and barcode information, was obtained. For the TCR, only cells with one productive TCR α-chain (TRA) and one productive TCR β-chain (TRB) were kept for further analysis. Each unique TRA(s)-TRB(s) pair was defined as a clonotype. If one clonotype was present in at least two cells, cells harboring this clonotype were considered to be clonal and the number of cells with such pairs indicated the degree of clonality of the clonotype.TCR clones that exist in both MWA + 2DG and MWA + PBS were defined as "Shared clones".

## CITE-seq analysis

CellRanger (7.0.0) to align reads to the genome (mm10-2020-A) with default settings. Thresholds were determined based on the density distribution of Antibody-derived tags (ADT) from CITE-seq in the cells, with log2(1 + UMI) values greater than 7.83 defined as positive cells and less than 7.83 defined as negative cells.

## Trajectory analysis

Monocle 2.0 package (v 2.10.0) was used to analyse single-cell trajectories. We used the differentially expressed genes in $CD8_{Naive}$ cells to sort cells in pseudotime order.

## TDLN and isolated CD8+T cells bulk RNA-seq library construction

Total RNA was extracted from frozen TDLNs or isolated CD8+T cells using Trizol reagent (Thermo Fisher, USA) following the standard procedure. The purity and quantity of the total RNA were evaluated by Bioanalyzer 2100 and RNA 6000 Nano LabChip Kit (Agilent, USA). Samples with RIN number >7.0 were used for further analysis. Dynabeads Oligo (dT) (Thermo Fisher, USA) were used to obtain purified poly(A) mRNA from total RNA (1 μg), and the mRNA was purified for two rounds. Then, the mRNA was fragmented into small pieces under elevated temperatures by divalent cations. According to the protocol for the mRNA-seq sample preparation kit (Illumina, USA), The cleaved RNA fragments were reverse-transcribed to generate the ultimate cDNA library. The average insert size for the final cDNA library was 300 bp (±50 bp). The paired-end sequencing was performed on an Illumina sequence platform (LC Science, Hangzhou, China).

## Bulk RNA-seq data analysis

For quality control, the sequence was verified by fastp software (https://github.com/OpenGene/fastp). HISAT2 was used to map reads to the *mus musculus*. StringTie (https://ccb.jhu.edu/software/stringtie) was used to assemble the mapped reads of each sample. To estimate the expression levels of all transcripts, StringTie and ballgown (http://www.bioconductor.org/packages/release/bioc/html/ballgown.html) were used after the final transcriptome was generated. FPKM (fragment per kilobase of transcript per million mapped reads) value was calculated by StringTie and Ballgown to perform mRNAs expression abundance.

## Immune cell composition analysis

For immune cell composition prediction in TDLNs, CIBERSORTx (https://cibersortx.stanford.edu/) was used to analyze standard RNA-seq expression quantification metrics. The reference marker gene expression profile (GEP) matrix was assigned with a widely used mouse immune cell signature matrix[69]. During the analysis, quantile normalization was disabled, and 1000 permutations were used to calculate P values.

## Differentially expressed genes (DEGs) and GO Enrichment Analysis

We determined the differentially expressed genes (DEGs) of RNA seq by R package limma (v3.48.3). DEGs were selected with log(fold change) higher than 0.25 or lower than −0.25 and statistical significance ($P < 0.05$). Differentially expressed genes were then subjected to enrichment analysis of GO functions by R package clusterProfiler (v 4.0.5).

## Th1 and CD8+T cell function and CD8+T cell memory scores and estimated transcription factor analyses

Mouse gene sets (m5.go.v2022.1.Mm.symbols) of the function of Th1 and CD8+T cells were collected from the MsigDB database (http://www.gsea-msigdb.org/) and the Gene Ontology database (http://geneontology.org/). The scores of Th1 and CD8+T cell function were then calculated by R package GSVA (v 1.40.1) based on the gene sets. A memory CD8+T cell signature consisting of 66 genes expressed uniquely in CD8+ memory T cells was also applied[32]. The transcription factor was estimated by the algorithm Lisa (http://lisa.cistrome.org/)[41].

## Flow cytometry

For mouse TDLN or spleen flow cytometry, single cells of TDLN or single splenocytes were used. Peripheral blood of mice was treated with an erythrocyte-lysing reagent to remove red blood cells, and then the single cells were used for flow cytometry. Human PBMCs were resuspended with PBS after culturing in vitro for flow cytometry. The cells were stained with Zombie Aqua™ Fixable Viability Kit (BioLegend, Cat# 423101, 1:1000) at 25 °C for 15 min, incubated with a blocking monoclonal antibody to CD16/32 (BioLegend, Cat# 156603, 1:50; Cat# 422301, 1:20).

For surface staining of cells, cells were stained with fluorescence-conjugated antibodies in flow cytometry staining buffer (Biolegend, USA) for 30 min at 4 °C. Surface-stained cells were fixed by fixation Buffer (Biolegend, USA) for intracellular protein staining. Then the cells were permeabilized by Intracellular Staining Permeabilization Wash Buffer (Biolegend, USA) following the manufacturer's protocol. Permeabilized cells were inoculated with staining antibodies in a Permeabilization Wash Buffer to stain intracellular protein. The following fluorescence-conjugated antibodies (Biolegend, MBL and Proteintech) were used: Brilliant Violet 421™ anti-mouse F4/80 Antibody (Biolegend, Cat# 123131, 1:20), PerCP/Cyanine5.5 anti-mouse CD64 (FcγRI) Antibody (Biolegend, Cat# 139307, 1:80), Pacific Blue™ anti-mouse CD45 Antibody (Biolegend, Cat# 103125, 1:200), Alexa Fluor® 700 anti-mouse CD45 Antibody (Biolegend, Cat# 103127, 1:200), FITC anti-mouse CD45 Antibody (Biolegend, Cat# 103107, 1:200), FITC anti-mouse CD8 Antibody (MBL, Cat# K0227-4, 1:10), T-Select H-2Ld MuLv gp70 Tetramer-SPSYVYHQF-PE (MBL, Cat# TS-M521-1, 1:10), FITC anti-mouse CD3 Antibody (Biolegend, Cat# 100204, 1:50), PE anti-mouse I-A/I-E Antibody (Biolegend, Cat# 107607, 1:80), PerCP anti-mouse CD3ε Antibody (Biolegend, Cat# 100326, 1:20), APC anti-mouse CD8a Antibody (Biolegend, Cat# 100711, 1:80), Alexa Fluor® 700 anti-mouse/human CD44 Antibody (Biolegend, Cat# 103025, 1:200), Pacific Blue™ anti-mouse CD8a Antibody (Biolegend, Cat# 100725, 1:200), Pacific Blue™ Annexin V (Biolegend, Cat# 640917, 1:20), PE/Cyanine7 anti-mouse CD103 Antibody (Biolegend, Cat# 121425, 1:30), Brilliant Violet 421™ anti-mouse CD4 Antibody (Biolegend, Cat# 100443, 1:150), APC/Fire™ 750 anti-mouse CD4 Antibody (Biolegend, Cat# 100460, 1:80), PE anti-mouse/human CD44 Antibody (Biolegend, Cat# 103007, 1:80), APC anti-mouse CD62L Antibody (Biolegend, Cat# 104411, 1:80), APC anti-mouse CD25 Antibody (Biolegend, Cat# 102011, 1:80), APC anti-mouse CD11c Antibody (Biolegend, Cat# 117309, 1:80), PE/Cyanine7 anti-mouse CD25 Antibody (Biolegend, Cat# 101915, 1:40), PE anti-mouse CD183 (CXCR3) Antibody (Biolegend, Cat# 126505, 1:80), PE anti-mouse FOXP3 Antibody (Biolegend, Cat# 126404, 1:20), APC/Cyanine7 anti-mouse CD69 Antibody (Biolegend, Cat# 104525, 1:80), APC/Cyanine7 anti-mouse/human CD11b Antibody (Biolegend, Cat# 101225, 1:80), PE anti-human CD197 (CCR7) Antibody (Biolegend, Cat# 353203, 1:20), APC anti-human CD45RA Antibody (Biolegend, Cat# 304111, 1:20), Pacific Blue™ anti-human CD8a Antibody (Biolegend, Cat# 301026, 1:50), APC/Cyanine7 anti-human CD4 Antibody (Biolegend, Cat# 317417, 1:20), FITC anti-human CD3 Antibody (Biolegend, Cat# 317306, 1:20), PerCP/Cyanine5.5 anti-human CD8a Antibody (Biolegend, Cat# 301031, 1:20), PerCP/Cyanine5.5 anti-human CD8 Antibody (Biolegend, Cat# 344709, 1:20), PE/Cyanine7 anti-human CD62L Antibody (Biolegend, Cat# 304821, 1:20), Brilliant Violet 605™ anti-human CD197 (CCR7) Antibody (Biolegend, Cat# 353223, 1:20), Brilliant Violet 421™ anti-human CD39 Antibody (Biolegend, Cat# 328213, 1:20), STAT1 Polyclonal antibody with R-PE−conjugated Goat Anti-Rabbit IgG(H + L) (Proteintech, Cat#10144-2-AP, 1:150; Cat#SA00008-2, 1:50). Cell data were acquired on the BD FACSCanto

cytometer (BD Biosciences) and were analyzed using FlowJo 10.8.1 software.

## Statistics and reproducibility

Medians and interquartile ranges or mean and standard deviation (SD) were used to describe continuous variables mentioned in the study. Proportions and frequencies were employed to describe the categorical variables mentioned.

The paired or unpaired t-test and Wilcoxon rank sum test were applied in single-cell RNA sequencing analysis to compare continuous variables. Pearson's chi-square and Fisher's exact tests were used to compare categorical variables. All statistics from high-throughput experiments were analyzed using R (v 4.1.1, https://www.r-project.org/).

For data from experiments in vitro or in vivo, two-tailed t-tests were employed for all comparisons between two groups. Ordinary one-way ANOVA with Tukey's multiple comparisons test was utilized to compare various groups, such as the memory subsets of T cells in the spleen and peripheral blood of mice. Using the log-rank test, the survival rate was analyzed with the Kaplan–Meier method.

All the experiments were repeated independently at least two times except single-cell RNA sequencing, RNA sequencing, and in-vitro experiments of clinical samples.

## Data availability

The previously published single-cell RNA sequencing data of humans used in this study are available in the Genome Sequence Archive for Human (GSA-Human) database under accession code HRA002043 (https://ngdc.cncb.ac.cn/gsa-human/). The raw sequencing data are available for non-commercial purposes under controlled access because of data privacy laws, and access can be obtained by request to the corresponding authors. The request will be passed within 1 week and then the users will be given a download link valid for 1 year to download the raw data. RNA-seq data of mice generated in this study have been deposited in the GEO datasets under accession code GSE245638 (https://www.ncbi.nlm.nih.gov/geo/). The single-cell RNA sequencing data of mice has been deposited in the Genome Sequence Archive in National Genomics Data Center, China National Center for Bioinformation / Beijing Institute of Genomics, Chinese Academy of Sciences (GSA) under accession code CRA014364 (https://ngdc.cncb.ac.cn/gsa). Source data are provided with this paper. For any further questions, please contact Wenbin Zhou (zhouwenbin@njmu.edu.cn). Source data are provided with this paper.

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

## Acknowledgements

All schemes of experiment designs were created with https://www.biorender.com. We thank a lot for Yipeng Feng for the instruction in bioinformatics and Zhiqiang Xu for the assistance in the design of our study. This work was supported in part by the National Natural Science Foundation of China (81771953 W.Z., 82172683 S.W., and 82303710 M.Y.), the Natural Science Foundation of Jiangsu Province (BK20180108 W.Z., BK20230017 W.Z.), Jiangsu Province Capability Improvement Project through Science, Technology, and Education (Jiangsu Provincial Medical Key Discipline, ZDXK202222 S.W.), Jiangsu Province Excellent Postdoctoral Program (2023ZB006 M.Y.), a project funded by Jiangsu Provincial Science and Technology Department (BE2022807 S.W.), a project funded by Jiangsu Postgraduate Practice and Innovation Plan (JX10214028, X.T.), 2019 Changzhou Municipal Health and Health Commission Guidance Project Breast Benign Tumor Microwave Ablation (Hua Pan), a project funded by Jiangsu province college students' practice innovation training programs(202210312026Z W.Z.) and a project funded by the Priority Academic Program Development of Jiangsu Higher Education Institutions (W.Z.).

## Author contributions

S.W. and W.Z. conceived, designed, and supervised the study. X.T., X.M., P.L., M.Y., Hua Pan, and F.B. performed the experiments, collected the data, and drafted the manuscript. J.W., M.L., and K.Z. performed the statistical analysis. Hongwei Peng, Q.D., N.C., W.Q., and Hong Pan edited the manuscript. All authors approved the final version of the manuscript.

## Competing interests

The authors declare no competing interests.

## Additional information

[1]Department of Breast Surgery, Department of General Surgery, The First Affiliated Hospital with Nanjing Medical University, 300 Guangzhou Road, 210029 Nanjing, China. [2]Jiangsu Key Lab of Cancer Biomarkers, Prevention and Treatment, Jiangsu Collaborative Innovation Center For Cancer Personalized Medicine, School of Public Health, Nanjing Medical University, Nanjing 211166, China. [3]Department of General Surgery, Liyang Branch of Jiangsu Provincial People's Hospital, 70 Jianshe West Road, 213399 Liyang, China. [4]Department of Immunology, Nanjing Medical University, Nanjing 211166, China. [5]Department of Rheumatology and Immunology, The First Affiliated Hospital with Nanjing Medical University, 300 Guangzhou Road, 210029 Nanjing, China. [6]Pancreatic Center & Department of General Surgery, The First Affiliated Hospital with Nanjing Medical University, Nanjing 210029 Jiangsu, China. [7]Pancreas Institute of Nanjing Medical University, Nanjing 210029 Jiangsu, China. [8]The First Clinical Medical College of Nanjing Medical University, Nanjing 210029 Jiangsu, China. [9]These authors contributed equally: Xinyu Tang, Xinrui Mao, Peiwen Ling, Muxin Yu, Hua Pan. ✉e-mail: shwang@njmu.edu.cn; zhouwenbin@njmu.edu.cn

