## [Peer Review File · Nature Communications]

Glycolysis inhibition induces anti-tumor central memory CD8+ T cell differentiation upon combination with microwave ablation therapyREVIEWER COMMENTS

Reviewer #1 (Remarks to the Author):

Tang et al describe how microwave ablation of human tumors induces a glycolytic gene signature in peripheral blood CD8 T cells. They then hypothesise that this might hamper efficient memory T cell differentiation and therefore treat mice following MWA with the glycolysis inhibitor 2DG. This increases memory differentiation and protects mice against tumor rechallenges in multiple models. While the preclinical therapeutic evidence to combine MWA with glycolysis inhibition is very interesting, the proposed model and mechanism contains several flaws. My most important critique is to which extent the induction of memory differentiation upon MWA + 2DG is due to a T cell intrinsic effect. Only systemic administration of 2DG is used, which makes sense for clinical translation, but does not allow to discern potential non-T cell effects that can induce memory. The phenotype of the T cells following MWA + 2DG should be better described to understand what mechanism is at play and how the investigators could genetically interfere with the process to provide experimental evidence of the proposed hypothesis. Besides that, I have several other concerns about data interpretation and experimental design:

1) The authors only use IL7R and EOMES as memory markers. This is not sufficiently convincing of a memory signature. Furthermore, EOMES can also be a marker of T cell exhaustion. The increase in KLRG1 might indicate a mixed phenotype. Thus, the current data does not support the conclusion of the authors that "the memory effect of CD8 T cells after MWA increased". A more complete panel of effector versus memory markers should be tested.

2) Flow cytometry experiments and data presentation: if dead cells were not excluded from flow cytometry data, all flow cytometry data would be problematic. Furthermore, the authors look at percentages, "ratios", of CD3, CD4 and CD8 T cells in the spleen and blood of mice. In all figures, absolute numbers should be given instead and expressed as number/microliter blood or gram spleen. The given percentages of Tem and Tcm are incredibly low and not matching the gating strategy shown in Supplementary Fig 3a and the representative image Fig 3d and 3i, and several following figures.

Sometimes it is not possible to understand what the parental population is when giving percentages, because gating strategy is not always described.

3) In vivo experiments seem to be only performed once. Data would be scientifically more convincing if 2 independent experiments show similar results. I suppose the Y-axis in Fig 4d is cubic mm? In this case, the MWA + PBS group in Fig 4d has a 10-fold lower tumor burden as the same experimental group of Fig 2e. Furthermore, the experiment in Fig2 resulted in a 100% regrowth in MWA+PBS as compared to only 50% in Fig4. This demonstrates large interexperimental differences in tumor growth and warrants for independent repeats of each experiment.

4) The hypothesis of 2DG enhancing memory in the peripheral blood and not in the TDLN is confusing and supported by conflicting data. In fig 6h, the authors show that T cells from the TDLN can be differentiated to Tcm by culture with 2DG. Why would this not happen in vivo? Spleens of MWA+2DG-treated mice show increased proportions of Tcm. Furthermore, it is surprising that 2DG only induces Tcm in MWA peripheral blood T cells, and not untreated T cells, since 2DG has been reported to generally induce a memory phenotype in activated T cells. Indeed, the authors even show this in the liver patient samples in Fig 7e-f.

5) The description of the data on Fig 7b and 7c is not correct. The statistical test comparing NC with 2DG after MWA is not significant and no statistical information is given for the comparison between NC and 2DG in the "before MWA" group as well. What about CD62L expression in human T cells, as that marker was used for all mouse characterization and is also an established marker for human memory T cells.

minor:

1) The authors often use the term "memory effect". This is confusing, because the authors rather

refer to phenotypic marker on T cells, not on a functional effect. I would suggest to use the term "memory phenotype".

2) Please show the representative graph for OVA control stimulation in Fig 4f as well. The IFN γ signal seems only marginally higher than the negative population. Antigen-specific T cells usually express much more IFN γ when restimulated with peptide in vitro.

3) It is not clear which markers were used to discriminate Th1 cells by flow cytometry.

Reviewer #2 (Remarks to the Author):

The manuscript by Tang X. et al. describes the potential of the combination treatment with microwave ablation (MWA) and glycolysis inhibition by 2DG. Metabolic control of immune cells, such as effector T cells, becomes an important option to augment antitumor immunity. Since effector T cells metabolically compete with tumor cells to utilize glycolysis for their activation, proliferation, and survival, it is crucial to develop a metabolic environment favorable for antitumor immune cells for establishing effective antitumor immunity. Yet, any reagents have not been successfully translated into the clinic. The authors addressed the issue by focusing on MWA in combination with glycolysis inhibition by 2DG. They examined that metabolic status of peripheral CD8 $^+$ T cells and found that glycolysis became dominant, although OXPHOS, which is essential for memory T cells, was decreased after MWA. To modulate the metabolic status, they employed glycolysis inhibition by 2DG after MWA, and the combination treatment was tested in several murine tumor models (4T1, B16F10, and MC38). The combination with MWA and 2DG exhibited a long-lasting antitumor effect. The antitumor effect was attributed to the increase of CD8 $^+$ central-memory T cells. While the differentiation of CD8 $^+$ central-memory T cells was observed in tumor-draining lymph nodes, further enhancement was occurred in peripheral blood. In humans, CD8 $^+$ T cells from peripheral blood of patients with breast cancer or liver cancer who received MWA predominantly differentiated into CD8 $^+$ central-memory T cells. Then, the authors proposed that the combination treatment with MWA and glycolysis inhibition would be a novel strategy for effective cancer therapy.

General comments;

Analyzing combination treatment with MWA and metabolic modulation is interesting. However, this study is descriptive, and no mechanistic insight was provided. In addition, the impact of this study is severely impaired since there are critical problems with the experimental design and interpretation of the data, and several points were not adequately addressed. While the authors firstly examined immune cells in peripheral blood before and after MWA by single cell-RNAseq, no information of clinical outcomes of the patients examined was provided. Additionally, CD8 $^+$ T cells were separated in GZMH $^+$ and GZMK $^+$ without any rationales. More importantly, gene expression should be compared with other treatment methods such as surgery, otherwise they cannot conclude the potential antigen release by MWA.

Second, the authors employed rechallenge of tumor cells. However, in general, it is well known that concomitant immunity is observed in tumor bearing mice. Therefore, even in control mice, growth of rechallenged tumor cells should be inhibited. The authors need to explain this discrepancy.

Another important issue is whether glycolysis inhibition was suitable as an effective cancer immunotherapy. While OXPHOS is important for memory T cells as the authors indicated, glucose deprivation is harmful for the activation of effector T cells (PMID: 18792400), thereby, these memory T cells should also require glycolysis for their activation upon the exposure to the cognate antigens. Therefore, it must be important to consider the metabolic dependency based on each phase of antitumor immune responses.

Lastly, the authors imply the advantage of MWA for tumor antigen release. However, it has been already reported that the combination of MWA and CART cell therapy show an augmented antitumor efficacy (PMID: 36261437). Therefore, this manuscript at the current form lacks novelty and several important data, and thus will not represent a major advance in the field.

Specific points;

1. In Figure 1, glycolysis and OXPHOS were compared. However, neither glycolysis nor OXPHOS were elevated after MWA. Therefore, this comparison may not bring any conclusions.
2. In Figure 2, the combination with MWA and 2DG was examined. The authors only focused on CD8+ central-memory T cells. It has been reported that regulatory T (Treg) cells also can use OXPHOS for their survival and activation (PMID: 32640259). Did Treg cells exhibit any changes after MWA + 2DG treatment? In addition, why did 2DG treatment fail to show direct effects on tumor growth despite of the Warburg effect of tumor cells?
3. In Figure 3, the authors examined changes of immunological landscape after MWA + 2DG treatment. While the authors examined tumor-draining lymph nodes and peripheral blood, analysis of the tumor microenvironment is highly appreciated. In addition, as they argue the importance of antigen release after MWA, antigen-presenting cells such as dendritic cells should also be explored. Moreover, effector vs memory phenotypes of CD8+ T cells need to be examined in tumor antigen (AH1)-specific CD8+ T cells.
4. In Figure 4g, induction of tumor antigen-specific CD8+ T cells was lower in MWA alone group compared to control and 2DG treatment group. The authors need to explain this impairment of tumor antigen-specific CD8+ T cell induction in 2DG treatment group.
5. In Figure 5, the authors employed bulk RNA-seq for tumor-draining lymph nodes. Considering the importance for accurately examine CD8+ central-memory T cells in this study, single cell-RNAseq would strengthen their conclusion.
6. In Figure 6, the importance of peripheral blood rather than tumor-draining lymph nodes for CD8+ central-memory T cell differentiation is argued. However, the data only shows an expansion of CD8+ central-memory T cells in in vitro culture. As CD8+ central-memory T cell population was increased in tumor-draining lymph nodes, it is difficult to exclude the contribution of tumor-draining lymph nodes for CD8+ central-memory T cell differentiation. Cell tracing experiments should be necessary to confirm their conclusion.
7. In Figure 7, the authors examined human samples after MWA treatment. As there are no information of clinical outcomes, it could be said cherry picking.

Reviewer #3 (Remarks to the Author):

This study investigated minimally invasive thermal therapy as an approach to treat solid tumors. The authors focused on microwave ablation (MWA) of breast cancer and its effects on peripheral CD8+ T cells. They found that MWA slightly increased the memory effect of CD8+ T cells and the related metabolism was unfavorable to memory formation. However, inhibition of glycolysis after MWA enhanced CD8+ memory T cell development and resulted in persistent anti-tumor effects. The authors emphasized the tumor-draining lymph node (TDLN) as the site for this enhancement, but the draining LNs have been well established for their roles in Tcm generation. Also, use of LN resection is a very rough method. Although the study showed a new strategy of combining local ablation and glycolysis inhibition with a possible treatment of solid tumors, the underlying mechanism was less elucidated. Serious concerns and limitations were raised regarding the animal experiments.

1. 4T1 is not an ideal model for tumor immunology study due to its low immunogenicity. Previous studies have shown that absence or a very low number of CD8+ T cells were present in 4T1 tumor, making it generally regarded as a "cold tumor." It is recommended to compare local T cell infiltration before and after MWA. MMTV-PyMT mice are a spontaneous breast cancer model, closer to clinical breast cancer, and the author should use this model to repeat the experiments.

2. 2DG (2-deoxy-D-glucose) is a glycolysis inhibitor that preferentially targets tumor cells. However, the authors showed that 2DG had no effect on tumor cells (e.g., Fig. 4B), and only affects T cells, which raises doubts. Previous studies have shown that T cells express high levels of MDR1, which may pump the drug out and facilitate tumor cells to take up the drug.

3. The mechanistic experiments are insufficient. The inclusion of 2DG possibly accelerates apoptosis of memory CD8⁺ T cells. It is crucial for the authors to address this concern and carefully evaluate the potential impact of 2DG on memory T cell viability in their experiments. What is the impact of 2DG on fatty acid metabolism, glutaminolysis, glycogen metabolism and other metabolic pathways upon glycolysis inhibition, which should be analyzed.

4. There are serious concerns with the analysis of memory T cells presented in Figure 3. Analyzing memory T cells should be specific, such as using tetramers or CD45.1-OT1 to distinguish between specific and bystander memory T cells, rather than analyzing all T cells collectively. The analysis should be more targeted and focused to accurately evaluate and understand the presence and function of memory T cells. It is important for the authors to address these concerns.

5. For Figure 7, the authors should use tetramers to analyze tumor-specific T cells. For patients, infections can significantly influence the experimental results. Therefore, there are concerns regarding this part of the results.

Reviewer #1 (Remarks to the Author):

Tang et al describe how microwave ablation of human tumors induces a glycolytic gene signature in peripheral blood CD8 T cells. They then hypothesise that this might hamper efficient memory T cell differentiation and therefore treat mice following MWA with the glycolysis inhibitor 2DG. This increases memory differentiation and protects mice against tumor rechallenges in multiple models. While the preclinical therapeutic evidence to combine MWA with glycolysis inhibition is very interesting, the proposed model and mechanism contains several flaws. My most important critique is to which extent the induction of memory differentiation upon MWA + 2DG is due to a T cell intrinsic effect. Only systemic administration of 2DG is used, which makes sense for clinical translation, but does not allow to discern potential non-T cell effects that can induce memory. The phenotype of the T cells following MWA + 2DG should be better described to understand what mechanism is at play and how the investigators could genetically interfere with the process to provide experimental evidence of the proposed hypothesis. Besides that, I have several other concerns about data interpretation and experimental design:

Response: We appreciate your comments. Our results indicated that the protective immune responses against tumor rechallenge relied on CD8+T cells. In our ex-vivo experiments, 2DG could promote the memory phenotype of CD8+T cells after MWA, which correlated with the previous intrinsic effect of 2DG to induce memory CD8+T cells (J Clin Invest, 123(10):4479-88, PMID: 24091329), validating the T cell-intrinsic effect of 2DG. As you mentioned, systemic administration of 2DG does not allow to discern potential non-T cell effects that can induce memory. In our in-vivo experiments, percentages of cDC1 and th1 cells in TDLN were increased in the combination group, which might be upstream of the memory differentiation of CD8+T cells. The immune responses induced by the combination therapy were inextricably interwoven. It was hard to determine the extent of the intrinsic effect of 2DG. We designed the systemic administration of 2DG to make it easier for clinical translation. The extent of the intrinsic effect of 2DG didn't influence the clinical translation of this strategy of treatment because we focused mainly on the clinical translation. Thanks again for your understanding.

In our study, we focused more on the therapeutic effect of the combination therapy, but we also performed RNA-seq on the CD8+T cells in spleens isolated from the MWA and combination group. Genetically, CD8+T cells exhibited elevated transcription factor Stat1 enriched in the MWA+2DG group compared with the MWA+PBS group (fig 6l). An additional experiment was also conducted for validation of the important role of STAT1 signaling in memory induction of the combined therapy. The results indicated that inhibition of STAT1 significantly inhibited memory formation of CD8+T cells in the MWA+2DG group (fig 6m). Thus, STAT1 in the CD8+T cells might be important for the memory induction of the combination therapy. (Page 29: line 518-524)

1) The authors only use IL7R and EOMES as memory markers. This is not sufficiently convincing of a memory signature. Furthermore, EOMES can also be a marker of T cell exhaustion. The increase in KLRG1 might indicate a mixed phenotype. Thus, the current data does not support the conclusion of the authors that "the memory effect of CD8 T cells

after MWA increased". A more complete panel of effector versus memory markers should be tested.

Response: Thanks for your professional advice. Our previous data revealed that the memory phenotype of CD8+T cells might slightly increase after MWA. However, as you mentioned, the memory markers used previously might indicate a mixed phenotype. To accurately evaluate the memory phenotype after MWA, we re-analyzed our single-cell RNA data with widely used memory markers (IL7R, fig 1d-e). Moreover, a memory CD8+T cell signature consisting of 66 memory genes expressed uniquely in CD8+ memory T cells was then applied (fig 1f, Proc Natl Acad Sci U S A, 101(48):16885-90, PMID:15548615). The memory CD8+T cell score was also slightly increased as IL7R did. Our results indicated the memory phenotype slightly increased after MWA. We have replaced our previous data in the result part. Thanks again for your careful consideration. (Page 6: line 111-113, line 122-123)

2) Flow cytometry experiments and data presentation: if dead cells were not excluded from flow cytometry data, all flow cytometry data would be problematic. Furthermore, the authors look at percentages, "ratios", of CD3, CD4, and CD8 T cells in the spleen and blood of mice. In all figures, absolute numbers should be given instead and expressed as number/microliter blood or gram spleen. The given percentages of Tem and Tcm are incredibly low and not matching the gating strategy shown in Supplementary Fig 3a and the representative image Fig 3d and 3i, and several following figures.

Sometimes it is not possible to understand what the parental population is when giving percentages, because gating strategy is not always described.

Response: We appreciate your comments. During our experiments, we measured the viabilities of all the samples with over 85% for flow cytometry. As you advised us to exclude the dead cells, we repeated the flow cytometry experiments with about 5% dead cells excluded which still showed a similar trend to our previous data. In Figure 7, because of the lack of cryo-preserved samples, we repeated our experiments in 3 samples. We have replaced the previous data. We also added the absolute numbers to our results and similar results were obtained. We are sorry that we didn't make the parental population clear in our manuscripts, we have added the parental population to all flow cytometry data.

3) In vivo experiments seem to be only performed once. Data would be scientifically more convincing if 2 independent experiments show similar results. I suppose the Y-axis in Fig 4d is cubic mm? In this case, the MWA + PBS group in Fig 4d has a 10-fold lower tumor burden as the same experimental group of Fig 2e. Furthermore, the experiment in Fig2 resulted in a 100% regrowth in MWA+PBS as compared to only 50% in Fig4. This demonstrates large interexperimental differences in tumor growth and warrants for independent repeats of each experiment.

Response: Thanks for your concern. We did 2 independent in-vivo experiments before our submission and we added "all the experiments were repeated twice independently" to the figure legends. We added the mm³ to the Y-axis in Figure 4d. Figures below are the results of 2 independent experiments of 4T1 tumors.

repeat of 2e

fig 2e

repeat of 2k

fig 2k

repeat of 4b

fig 4b

repeat of 4d

fig 4d

The differences in the tumor growth between different figures were because we used different numbers of tumor cells to rechallenge. We had provided the cell number in the figure legends in the previous manuscript as figures below. 5×10^5 tumor cells were used in fig 2e but only 1×10^5 tumor cells were used in fig4d. Because the experiment in fig 4d depleted CD8+T cells of mice, tumors might grow much faster in mice without CD8+T cells than in other mice. To adhere to the principles of animal ethics, fewer tumor cells were used in fig 4d.

a

229 (n = 4 per group). **d** Experimental design for (e and f). WT BALB/c mice implanted with 4T1 mammary tumor cells

13

230 were treated as in (a). 4T1 tumor cells ($n = 5 \times 10^5$) were re-challenged at the opposite flanks of mice from the 4

231 groups on day 35. **e** Tumor volume curves of mice from the 4 groups in (d) re-challenged with 4T1 tumor cells on

b

339 in (a) re-challenged with 4T1 tumor cells on day 35 (n = 5 per group). c Experimental design for (d). WT BALB/c
340 mice implanted with 4T1 tumor cells were treated as in (a). Mice in the MWA+2DG group were intraperitoneally
341 injected with anti-CD8a or rat IgG2a,κ q.4d from day 34 to day 50. Mice in the MWA group were intraperitoneally
342 injected with rat IgG2a,κ q.4d from day 34 to day 50. 4T1 tumor cells (n = 1 × 10⁵) were re-challenged at the opposite
343 flanks of mice from the 3 groups on day 35. d Tumor volume curves of the mice re-challenged with 4T1 tumor cells

Figure legend: The previous figure legends of Figure 2 (a) and Figure 4 (b).

4) The hypothesis of 2DG enhancing memory in the peripheral blood and not in the TDLN is confusing and supported by conflicting data. In fig 6h, the authors show that T cells from the TDLN can be differentiated to Tcm by culture with 2DG. Why would this not happen in vivo? Splens of MWA+2DG-treated mice show increased proportions of Tcm. Furthermore, it is surprising that 2DG only induces Tcm in MWA peripheral blood T cells, and not untreated T cells, since 2DG has been reported to generally induce a memory phenotype in activated T cells. Indeed, the authors even show this in the liver patient samples in Fig 7e-f.

Response: Thanks for the helpful concerns. Theoretically, following activation within secondary lymphoid tissue, CD8+T cells will migrate into the periphery (Nat Rev Immunol, 16(3):193-201, PMID: 26852928). Thus, T cells with anti-CD3 and anti-CD28 isolated from the TDLNs would migrate out of the TDLN into the blood if they were in vivo. This explained why T cells isolated from the TDLN could differentiate into CD44^{hi}CD62L⁺CD8+T cells (CD8+Tcm) induced by 2DG. Moreover, Tcm cells migrate primarily from the blood into secondary lymphoid organs (Janeway's immunology, 9th edition), which also supports the possibility of differentiation of Tcm in the peripheral blood. Our ex-vivo experiments indicated that CD8+T cells from the peripheral blood could differentiate into CD44^{hi}CD62L⁺CD8+T cells induced by 2DG (fig 6g). We performed additional experiments to further validate our hypothesis. FTY720 blocking lymphocytes egress from the lymph node was used in our experiments (fig 6i-j). Without lymphocytes exiting TDLN, 2DG could not induce memory differentiation of tumor-specific CD8+T cells in the TDLN (fig 6i-j). These data further indicated that 2DG might promote CD44^{hi}CD62L⁺CD8+T(CD8+Tcm) cell differentiation in peripheral blood. We have modified the descriptions of the results of our ex-vivo experiments in fig6. Data from in-vivo experiments with FTY720 were also added to the results. (Page 28: line 497-502)

The metastases of 4T1 tumor cells are efficient and usually occur within 2-4 weeks after 4T1 inoculation, thus the progression of 4T1 is very rapid. In our ex-vivo experiments of mice, we did MWA or not on day 10 and blood was harvested on day 15. The local and systemic immune responses in the non-treatment group would accumulate much more inhibitory immune responses than those in the MWA group because of the rapid progress of the primary tumor (Clin Exp Metastasis, 31(2):185-98, PMID: 24096737). The induction

of memory relied on the activation of CD8+T cells, but the CD8+T cells from the non-treatment group were much less activated than those from the MWA group in mice (Figure below). We completely agree with you that 2DG has been reported to generally induce a memory phenotype in activated T cells. Thus, we repeated our experiments in all previous clinical cohorts by excluding dead cells (supplementary fig 7b), and a new cohort consisting of 9 consecutive patients with early-stage breast cancer (supplementary fig 7b) was included. Similar results were obtained. 2DG could induce CD8+Tcm cell differentiation in the samples before MWA too. Thank you again for your valuable comments, which have enabled us to improve our work. (Page 32: line 571-573)

Figure legend: WT BALB/c mice were implanted with 4T1 mammary tumor cells on day 0. Mice were treated with MWA or not on day 10. Peripheral blood was collected on day 15 for flow cytometry. Percentage of CD44hiCD8+T cells in the blood of mice from the 2 groups on day 15 (n = 3 per group). *P < 0.05, **P < 0.01, ***P < 0.001, ****P < 0.0001.

5) The description of the data on Fig 7b and 7c is not correct. The statistical test comparing NC with 2DG after MWA is not significant and no statistical information is given for the comparison between NC and 2DG in the “before MWA” group as well. What about CD62L expression in human T cells, as that marker was used for all mouse characterization and is also an established marker for human memory T cells.

Response: Thanks for your helpful advice. We have modified the description of our new results. P-values were also added to the figures (supplementary fig 7b). We also considered using CD62L as a memory marker before. However, in our cohorts, all samples were cryopreserved. It was well-known that PBMC cryopreservation could result in preferential loss of CD62L (Guidelines for the use of flow cytometry and cell sorting in immunological studies (second edition), PMID: 31633216), making CD62L unsuitable for use in our study. Moreover, CCR7 and CD45RA were suitable for immune monitoring of peripheral blood naïve/memory T cells by validation of a multi-center study (J Immunol Res, 2020:1938704, PMID: 32322591). Thus, we finally chose CCR7 as our memory marker in flow cytometry. We sincerely appreciate your understanding. (Page 32-33: line 566-582)

minor:

1) The authors often use the term “memory effect”. This is confusing, because the authors rather refer to phenotypic marker on T cells, not on a functional effect. I would suggest to use the term “memory phenotype”.

Response: Thank you for your suggestions regarding the use of terminology in our article. We agree that the term "memory effect" may be confusing since we are referring to a phenotypic marker on T cells rather than a functional effect. We have used the term "memory phenotype" in the revisions to improve clarity and accuracy.

2) Please show the representative graph for OVA control stimulation in Fig 4f as well. The IFN γ signal seems only marginally higher than the negative population. Antigen-specific T cells usually express much more IFN γ when restimulated with peptide in vitro.

Response: Thanks for your concern. The representative graph for OVA control was provided (Figure below). It is well-known that this experiment is an indirect method to determine tumor-specific T cells, whose results can be partially influenced by the viability, and status of T cells. The status of T cells in different groups was totally different such as more memory CD8+T cells in the MWA+2DG group. To further exclude the influence of different statuses of T cells among the 4 groups, we decided to use tetramer to directly evaluate the tumor-specific T cells. Consistently, more tumor specific CD8+T cells were found in the MWA+2DG group compared to other groups in the additional experiments and we have replaced the previous data with new data (fig 4g). (Page 19-20: line 355-366)

Figure legend: The representative flow plots of the peptide restimulation experiments.

3) It is not clear which markers were used to discriminate Th1 cells by flow cytometry.

Response: Thanks for your helpful advice. We have added the marker CXCR3 (Guidelines for the use of flow cytometry and cell sorting in immunological studies (second edition), PMID: 31633216; Exp Cell Res, 317(5):620-31, PMID: 21376175) to our results (fig 5n). The gating strategy was provided in the supplemental information (Supplementary fig5b).

Reviewer #2 (Remarks to the Author):

The manuscript by Tang X. et al. describes the potential of the combination treatment with microwave ablation (MWA) and glycolysis inhibition by 2DG. Metabolic control of immune cells, such as effector T cells, becomes an important option to augment antitumor immunity. Since effector T cells metabolically compete with tumor cells to utilize glycolysis for their activation, proliferation, and survival, it is crucial to develop a metabolic environment favorable for antitumor immune cells for establishing effective antitumor immunity. Yet, any reagents have not been successfully translated into the clinic. The authors addressed the issue by focusing on MWA in combination with glycolysis inhibition by 2DG. They examined that metabolic status of peripheral CD8⁺ T cells and found that glycolysis became dominant, although OXPHOS, which is essential for memory T cells, was decreased after MWA. To modulate the metabolic status, they employed glycolysis inhibition by 2DG after MWA, and the combination treatment was tested in several murine tumor models (4T1, B16F10, and MC38). The combination with MWA and 2DG exhibited a long-lasting antitumor effect. The antitumor effect was attributed to the increase of CD8⁺ central-memory T cells. While the differentiation of CD8⁺ central-memory T cells was observed in tumor-draining lymph nodes, further enhancement was occurred in peripheral blood. In humans, CD8⁺ T cells from peripheral blood of patients with breast cancer or liver cancer who received MWA predominantly differentiated into CD8⁺ central-memory T cells. Then, the authors proposed that the combination treatment with MWA and glycolysis inhibition would be a novel strategy for effective cancer therapy.

General comments;

Analyzing combination treatment with MWA and metabolic modulation is interesting. However, this study is descriptive, and no mechanistic insight was provided. In addition, the impact of this study is severely impaired since there are critical problems with the experimental design and interpretation of the data, and several points were not adequately addressed. While the authors firstly examined immune cells in peripheral blood before and after MWA by single cell-RNAseq, no information of clinical outcomes of the patients examined was provided. Additionally, CD8⁺ T cells were separated in GZMH⁺ and GZMK⁺ without any rationales. More importantly, gene expression should be compared with other treatment methods such as surgery, otherwise they cannot conclude the potential antigen release by MWA.

Response: Thanks for the helpful suggestions. Our single-cell RNA seq was performed in our phase I clinical trial (*Adv Sci (Weinh)*, 9(17): e2200033, PMID: 35403824) to evaluate the feasibility of MWA of patients with early-stage breast cancer. The clinical trial didn't focus on the long-term outcome and all enrolled patients are still alive now because all patients were treated with standard therapy 7 days after MWA. The short-term outcome of this study was complete ablation. All information about short-term outcomes and clinical characteristics can be found in our previous study (*Adv Sci (Weinh)*, 9(17): e2200033, PMID: 35403824). Seurat package was used for dimensionality reduction, clustering, and visualization, which was also described in our previous clinical trial (*Adv Sci (Weinh)*, 9(17): e2200033, PMID: 35403824). We have cited our previous clinical trial in our result and method parts.

As we mentioned in the limitation part, we couldn't accurately examine the released antigen after MWA because of a lack of known tumor antigens in breast cancer. However, others reported the tumor antigen release into the blood after ablation of other tumor types (Urol Oncol, 23(1):8-11, PMID: 15885576; Cancer, 107(1):149-53, PMID: 16736515). Indirectly, Damian E. Dupuy et al. conclude that by thermal ablation of tumors, the tumor antigens are released after necrosis drain to nearby lymph nodes, where they can stimulate immature DCs and naive T cells (Nat Rev Cancer, 14(3):199-208, PMID: 24561446). Consistently, our results validated the tumor necrosis and the elevated percentages and absolute numbers of DCs and tumor-specific CD8+T cells in the TDLN after MWA (Figure below). Other studies also indicated the in-situ antigen release after thermal ablation (Cancer Res, 64(11):4024-9, PMID: 15173017; Nat Commun, 13(1):6203, PMID: 36261437; Cancer Immunol Immunother, 66(2):247-258, PMID: 27585790; J Immunother Cancer, 5(1):78, PMID: 29037259; Clin Cancer Res, 22(5):1173-1184, PMID: 26933175). Our previous data on the expansion of TCR β repertoire and elevated TCR diversities induced by MWA (J Immunother Cancer, 9(4):e002343, PMID:33795388; Adv Sci (Weinh), 9(17):e2200033, PMID: 35403824) also indicated the antigen release after MWA of breast cancer.

Figure legend: WT BALB/c mice were implanted with 4T1 mammary tumor cells on day 0. Mice were treated with MWA or surgery (baseline) on day 10. TDLNs were collected on

day 15 for flow cytometry. Percentages and absolute numbers of DCs (a,b) and tetramer+CD8+T(c,d) cells in TDLNs of mice from the 2 groups on day 15 (n = 3 per group). *P < 0.05, **P < 0.01, ***P < 0.001, ****P < 0.0001.

Figure legend: WT C57BL/6 mice were implanted with Py8119 mammary tumor cells on day 0. Mice were treated with MWA on day 10. The ablated tumors were collected on 0D, 3D, and 7D after MWA. Representative H-E stainings (a-c) were shown. Scale bar, 20um.

Second, the authors employed rechallenge of tumor cells. However, in general, it is well known that concomitant immunity is observed in tumor bearing mice. Therefore, even in control mice, growth of rechallenged tumor cells should be inhibited. The authors need to explain this discrepancy.

Response: Thanks for the comment. We agree with you that concomitant immunity is observed in tumor-bearing mice. However, Immune suppression is also clearly observed in patients with cancer and tumor-bearing animals. Although cancers often elicit a vigorous immune response during the early part of their growth, the immune response is soon downregulated, permitting progressive tumor growth (Nat Rev Cancer, 5(4):263-74, PMID: 15776005; J Immunol, 166(1):678-89, PMID: 11123353). Late-stage tumors have been reported to accumulate immunosuppressive factors such as MDSCs systemically (Signal Transduct Target Ther, 6(1):362, PMID: 34620838; Immunity, 54(5):875-884, PMID: 33979585; J Immunol, 166(1):678-89, PMID: 11123353). We did our rechallenge on day 35 (late-stage) when the immune responses of mice in the control and 2DG groups were downregulated. Thus, the growth of the rechallenged tumor cells could be permitted by immunosuppression.

Another important issue is whether glycolysis inhibition was suitable as an effective cancer immunotherapy. While OXPHOS is important for memory T cells as the authors indicated, glucose deprivation is harmful for the activation of effector T cells (PMID: 18792400), thereby, these memory T cells should also require glycolysis for their activation upon the exposure to the cognate antigens. Therefore, it must be important to consider the metabolic dependency based on each phase of antitumor immune responses.

Response: Thanks for your concern. This is a very meaningful comment. We agree with your concern about metabolic dependency based on each phase of antitumor immune responses. During the memory-induction phase, both the activated CD8+T cells after MWA

and systemic 2DG may exert anti-tumor effects. However, as you mentioned, 2DG might also inhibit the anti-tumor effect of CD8+T cells when they are exposed to the cognate antigens. Thus, we isolated CD8+T cells from the spleens of mice in MWA+2DG and MWA+PBS groups 7 days after the MWA to coculture with tumor cells. We added 2DG or PBS into the coculture medium respectively to evaluate the overall anti-tumor effects. The results demonstrated that during the memory-induction phase, the presence of 2DG didn't reduce the overall anti-tumor responses (fig 3i-j). After the memory induction phase, the memory CD8+T cells can provide long-term anti-tumor effects systemically and rapidly expand when they recognize the tumor cells without the influence of 2DG. (Page 16: line 286-293)

Clinically, complete tumor ablation has been used as a standard local therapy for several solid tumors. The complete tumor ablation is locally curative as surgery. However, tumor metastasis or recurrence still happens in a few patients. Our study focused more on the long-term anti-tumor effect instead of the short-term effect after complete local ablation of early-stage tumors. Thanks again for the important comment which helped us improve our manuscript a lot. We have discussed this issue in our discussion section and added the new data to our result section.

Lastly, the authors imply the advantage of MWA for tumor antigen release. However, it has been already reported that the combination of MWA and CART cell therapy show an augmented antitumor efficacy (PMID: 36261437). Therefore, this manuscript at the current form lacks novelty and several important data, and thus will not represent a major advance in the field.

Response: Thank you very much for your comments. We have cited this article and discussed the MWA+CART therapy in the discussion section. Undoubtedly, MWA and CART cell therapy is a major advance in the field. The study applied partial ablation of the primary lung tumor, validating the immune-modulatory effect of partial MWA in lung tumors. Thus, the study focused on the partial ablation of lung tumors which might simulate the local therapy for late-stage lung tumors in the clinical practice because the early-stage tumors are always ablated completely. However, our combination therapy applied complete ablation of the primary tumor, which simulated the early-stage tumor in clinical. We mainly focused on reducing the metastasis or recurrence after complete ablation of the early-stage tumor, which is always used as an effective local therapy in patients with early-stage tumors. The complete tumor ablation is locally curative as surgery. However, recurrence or metastasis still occurs in patients. Short-term glycolysis inhibition enhanced the long-term immune responses to prevent recurrence and metastasis after complete MWA. Overall, the combination of complete MWA and 2DG might be a novel treatment for early-stage tumors, while MWA and CART cell therapy is an effective therapy for late-stage lung cancer.

Secondly, AXL (Nat Commun, 13(1):6203, PMID: 36261437) was an excellent tumor target for CART cells in lung cancer with very low expression in normal tissues. Thus, MWA+CART therapy can be an effective and safe therapy. In this combination therapy, MWA performed an immune-modulatory effect, and CART cells were applied to kill a specific cluster of tumor cells with known targets. However, in our study, complete tumor

ablation led to complete tumor clearance locally and promoted the release of varieties of tumor antigens even from tumors with high heterogeneity (Nat Rev Cancer, 14(3):199-208, PMID: 24561446). Many tumors exhibited heterogeneity between different patients and even within the same tumor (Nature, 618(7965):598-606, PMID: 37258682; J Clin Invest, 121(10):3786-8, PMID: 21965334; Cancer Lett, 379(2):191-7, PMID: 26213370). It might be hard to find a safe and effective target for CART therapy for tumor types with high heterogeneity. Thermal ablation of tumors promotes the release of tumor antigens. Thus, complete tumor ablation may not be limited to the tumor heterogeneity, which means complete tumor ablation might be an effective local therapy for many tumor types without targets such as breast cancer. 2DG could further enhance the long-term anti-tumor immunity induced by the local complete tumor ablation. The anti-tumor effect of the combination therapy was tested in several tumor models in our study. We believe that MWA+2DG could show a broad range of applications in different types of tumors.

Thirdly, previous clinical experience proved good tolerability of 2DG alone or 2DG combined with chemotherapy or radiotherapy in solid cancers (Prostate, 70(13):1388-94, PMID: 20687211; Strahlenther Onkol, 181(8):507-14, PMID: 16044218; Cancer Chemother Pharmacol, 71(2):523-30, PMID: 23228990). 2DG also exploits viral-induced increased glucose uptake and metabolism by blocking glycolysis and interfering with N-linked glycosylation, which leads to emergency approval of 2DG in India to be used against SARS-CoV-2 (IUBMB Life, 73(10):1198-1204, PMID: 34418270). The safety of using 2DG was endorsed in patients with cancer. In our previous clinical trial (Radiology, 263(2):364-73, PMID: 22438362), Minor complications were encountered with MWA therapy. Slight thermal injuries to the skin and pectoralis major muscle, which proved reversible, were found in 3 of 41 cases. Thus, MWA+2DG may be a safe strategy for clinical translation.

Specific points;

1. In Figure 1, glycolysis and OXPHOS were compared. However, neither glycolysis nor OXPHOS were elevated after MWA. Therefore, this comparison may not bring any conclusions.

Response: Thanks for your helpful concern. Clinically, the peripheral CD8+T cells were weakly activated after complete MWA in patients with early-stage breast cancer (J Immunother Cancer, 9(4):e002343, PMID:33795388; Adv Sci (Weinh), 9(17):e2200033, PMID: 35403824), but recurrence or metastasis still occurs. Thus, there is an urgent need to enhance long-term immunity after complete MWA. As previously reported, activated CD8+T cells exhibit elevated rates of glycolytic activity while memory CD8+T cells preferentially rely on OXPHOS to support their energy demands (Cell Metab, 26(1):94-109, PMID: 28683298, Nat Rev Cancer, 20(9):516-531, PMID: 32632251; Immunol Rev, 283(1):213-231, PMID: 29664569). Thus, the two metabolic pathways (glycolysis and OXPHOS) are extremely important for the energy supply of T cells, which have been shown to play key roles in T-cell fate and longevity (Immunity, 45(5):1024-1037, PMID: 27836431; J Clin Invest, 123(10):4479-88, PMID: 24091329; Cell Metab, 26(1):94-109, PMID: 28683298).

To investigate the energy metabolism status of CD8+T cells before and after MWA, glycolysis score, and OXPHOS score were calculated in our single-cell RNA seq data (fig 1g-h). Delta score (D-value between two scores) was previously utilized to determine T-cell status (Cell, 186(6):1127-1143.e18, PMID: 36931243). We utilized the delta score (Glycolysis – OXPHOS) to evaluate the changes in metabolism before and after MWA (fig 1i). Our data suggested that CD8+T cells showed a trend of more reliance on glycolysis than on OXPHOS for energy supply after MWA (fig 1i). The energy metabolism status of CD8+T cells after MWA was unfavorable for memory T cells. Thus, glycolysis inhibition was chosen in our study to promote the memory phenotype of CD8+T cells after MWA. We are sorry that we didn't make this experiment design clear.

2. In Figure 2, the combination with MWA and 2DG was examined. The authors only focused on CD8+ central-memory T cells. It has been reported that regulatory T (Treg) cells also can use OXPHOS for their survival and activation (PMID: 32640259). Did Treg cells exhibit any changes after MWA + 2DG treatment? In addition, why did 2DG treatment fail to show direct effects on tumor growth despite of the Warburg effect of tumor cells?

Response: Thanks for your professional advice. We have tested the percentages of Treg among the four groups. No obvious difference was observed between the MWA+2DG and MWA+PBS groups (fig 3f). In our clinical settings, it took a short period to induce CD8+T cell memory. After the memory induction phase (day 10 – day 30), we did tumor rechallenge experiments to test the memory anti-tumor effect. However, the remaining 2DG in vivo might also influence the growth of rechallenged tumors. We therefore did the tumor rechallenge on day 35 (5 days after the last injection of 2DG) to exclude the anti-tumor effect of 2DG. There should be no remaining 2DG on day 35 because the plasma half-life of 2DG was only 48min (Int J Mol Sci, 21(1):234, PMID: 31905745). Moreover, we used UPLC-ESI-MS/MS to further validate that there was no remaining 2DG 5 days after the last injection of 2DG (fig 2d). (Page 15-16: line 279-282; Page 10: line 187-188)

3. In Figure 3, the authors examined changes of immunological landscape after MWA + 2DG treatment. While the authors examined tumor-draining lymph nodes and peripheral blood, analysis of the tumor microenvironment is highly appreciated. In addition, as they argue the importance of antigen release after MWA, antigen-presenting cells such as dendritic cells should also be explored. Moreover, effector vs memory phenotypes of CD8+ T cells need to be examined in tumor antigen (AH1)-specific CD8+ T cells.

Response: We appreciate your valuable advice, and it is very helpful. We are very sorry that the primary tumor cannot be obtained because we did complete ablation of the local tumor which was in line with the clinical practice, different from previous reported partial ablation research (Nat Commun, 13(1):6203, PMID: 36261437; J Transl Med, 20(1):433, PMID: 36180876; Adv Sci, 9(7):e2102182, PMID: 35037422). We had tried to discover the local immune infiltration in our previous studies (Cell Mol Immunol, 18(9):2153-2164, PMID: 32385362), which was too hard because of the complete ablation of the tumors. IHC staining of CD3 on day 7 after MWA also validated that there were no infiltrated T cells in the ablated tumor (figure below). Moreover, on day 35, the ablated tumor was absorbed so

that we could not harvest tumor tissues. As you advised, we explored the changes in DCs between the MWA+2DG and MWA+PBS groups. The percentages of DC, cDC1, and migratory cDC1 were increased in the MWA+2DG group compared with the MWA+PBS group (fig 5p-q). Moreover, we did additional experiments to exclude bystander T cells with AH1 tetramer, which exhibited similar results. (Page 24: line 437-441)

Figure legend: Figure legend: WT C57BL/6 mice were implanted with Py8119 mammary tumor cells on day 0. Mice were treated with MWA on day 10. The ablated tumors were collected on 0D, 3D, and 7D after MWA. Representative H-E staining (a-c) or IHC staining (c) sections were shown. Scale bar, 20um.

4. In Figure 4g, induction of tumor antigen-specific CD8+ T cells was lower in MWA alone group compared to control and 2DG treatment group. The authors need to explain this impairment of tumor antigen-specific CD8+ T cell induction in 2DG treatment group.

Response: We appreciate your concern. It is well-known that the peptide restimulation experiment was an indirect method to determine tumor-specific T cells, whose results can be partially influenced by the viability and status of T cells. The status of T cells in different groups of mouse models was totally different such as more memory status of CD8+T in the MWA+2DG group but more inhibitory status of CD8+T in the MWA+PBS group. Moreover, dead cells were not excluded in this experiment. Thus, the indirect experiment could not accurately determine the tumor-specific T cells. We decided to use tetramer to directly evaluate the tumor-specific T cells. Consistently, more tumor-specific CD8+T cells were found in the MWA+2DG group compared to other groups in the additional experiments and we have replaced the previous data with new data (fig 4g). (Page 19-20: line 355-362)

5. In Figure 5, the authors employed bulk RNA-seq for tumor-draining lymph nodes. Considering the importance for accurately examine CD8+ central-memory T cells in this study, single cell-RNAseq would strengthen their conclusion.

Response: Thanks for your suggestion. In our study, RNA-seq data was only an indication of the activation of CD8+T cells. We had performed flow cytometry to validate the results of RNA-seq. According to your suggestion, our results might be a little bit inaccurate because of not excluding the influence of other cells. Single-cell RNA seq is an accurate method to analyze functions of CD8+T-cell subsets. However, many studies failed to discern CD8+Tcm cells in their single-cell RNA seq data with only CD8+Tem or CD8+Tm annotated (Nat Commun, 13(1):6823, PMID: 36357424; Science, 374(6574):abe6474, PMID: 34914499). Moreover, bystander T cells cannot be excluded by single-cell RNA seq. Thus, to make our conclusion more convincing, we used tumor-antigen tetramer to exclude

bystander T cells, and similar results were obtained. We also isolated the CD8+T cells in the spleens of mice from the MWA+2DG and MWA+PBS groups on day 35 for RNA-seq. Consistently, the CD8+T memory score was elevated in the CD8+T cells from the MWA+2DG group compared with that from the MWA+PBS group (Supplementary fig 6b). Thanks again for your suggestion to make our results more convincing. (Page 28: line 513-515)

6. In Figure 6, the importance of peripheral blood rather than tumor-draining lymph nodes for CD8+ central-memory T cell differentiation is argued. However, the data only shows an expansion of CD8+ central-memory T cells in in vitro culture. As CD8+ central-memory T cell population was increased in tumor-draining lymph nodes, it is difficult to exclude the contribution of tumor-draining lymph nodes for CD8+ central-memory T cell differentiation. Cell tracing experiments should be necessary to confirm their conclusion.

Response: Thanks for your valuable advice. Theoretically, following activation within secondary lymphoid tissue, CD8+T cells will migrate into the peripheral blood (J Immunol, 172(8):4875-82, PMID: 15067066; Nat Rev Immunol, 16(3):193-201, PMID: 26852928). Thus, our ex-vivo experiments of TDLNs with anti-CD3 and anti-CD28 corresponded to the activated CD8+T cells that had migrated out of the TDLN into the blood. This explained why T cells isolated from the TDLN could differentiate into CD44^{hi}CD62L⁺CD8+T cells (CD8+Tcm) induced by 2DG. Moreover, Tcm cells migrate primarily from the blood into secondary lymphoid organs (Janeway's immunology, 9th edition), which supported the possibility of differentiation of Tcm in the peripheral blood. Our ex-vivo experiments indicated that CD8+T cells from the peripheral blood could differentiate into CD44^{hi}CD62L⁺CD8+T cells induced by 2DG (fig 6g). We performed additional experiments to further validate our hypothesis. By using FTY720 blocking lymphocytes egress from the TDLN, 2DG could not induce memory differentiation of tumor-specific CD8+T cells in the TDLN (fig 6i-j). These data further indicated that 2DG might promote CD44^{hi}CD62L⁺CD8+T(CD8+Tcm) cell differentiation in peripheral blood. We have modified the descriptions of the results of our ex-vivo experiments in fig6. Data from in-vivo experiments with FTY720 were also added to the results. (Page 28: line 497-502)

7. In Figure 7, the authors examined human samples after MWA treatment. As there are no information of clinical outcomes, it could be said cherry picking.

Response: Thanks for your professional concern. All patients enrolled in our study were from our phase I clinical trial (Adv Sci (Weinh), 9(17): e2200033, PMID: 35403824) to evaluate the feasibility of MWA of patients with early-stage breast cancer. The clinical trial didn't focus on the long-term outcome and all enrolled patients are still alive now because all patients were treated with standard therapy 7 days after MWA. The short-term outcome of this study was complete ablation. Our study aimed to propose the combination of MWA and 2DG as an effective therapy for solid tumors. We only validated the memory induction of 2DG after MWA in vitro. The best validation of the combined therapy should be the long-term outcomes of patients in vivo and we are going to conduct a clinical trial of MWA+2DG

therapy. We added the lack of long-term outcomes of the combination therapy to our limitation part. To make our results more convincing, 9 consecutive patients with early-stage breast cancer who received MWA in our phase II clinical trial (NCT04805736) were enrolled. PBMCs of the 9 consecutive patients were collected to repeat the experiment. Similar results were obtained (supplementary fig 7b), and we have added them to the result part. (Page 38: line 685-687; Page 32: line 571-573)

Reviewer #3 (Remarks to the Author):

This study investigated minimally invasive thermal therapy as an approach to treat solid tumors. The authors focused on microwave ablation (MWA) of breast cancer and its effects on peripheral CD8+ T cells. They found that MWA slightly increased the memory effect of CD8+ T cells and the related metabolism was unfavorable to memory formation. However, inhibition of glycolysis after MWA enhanced CD8+ memory T cell development and resulted in persistent anti-tumor effects. The authors emphasized the tumor-draining lymph node (TDLN) as the site for this enhancement, but the draining LNs have been well established for their roles in Tcm generation. Also, use of LN resection is a very rough method. Although the study showed a new strategy of combining local ablation and glycolysis inhibition with a possible treatment of solid tumors, the underlying mechanism was less elucidated. Serious concerns and limitations were raised regarding the animal experiments.

Response: Thanks for your concern. The TDLNs are considered common sites of metastasis in several tumor types such as breast cancer and we did LN resection in our experiments because TDLNs are always dissected in surgery for several tumor types. Thus, the resection of the TDLN in our experiment corresponded to the dissection of LNs in clinical practice. Others also used TDLN resection as an experimental method previously (JCI Insight, 3(23):e124507, PMID: 30518694; iScience, 23(5):101056, PMID: 32344378; Cancer Commun (Lond), 42(10):971-986, PMID: 35962977). We did lymph node resection gently with a sham surgery group to exclude the influence of operation techniques. Moreover, according to your advice, we did additional experiments by FTY720 to block T cells from migrating out of the TDLN to validate the importance of TDLN. Additional experiments were also done to further validate the site for the enhancement of memory.

1. 4T1 is not an ideal model for tumor immunology study due to its low immunogenicity. Previous studies have shown that absence or a very low number of CD8+ T cells were present in 4T1 tumor, making it generally regarded as a "cold tumor." It is recommended to compare local T cell infiltration before and after MWA. MMTV-PyMT mice are a spontaneous breast cancer model, closer to clinical breast cancer, and the author should use this model to repeat the experiments.

Response: Thanks for your advice and it is very helpful. We are very sorry that the primary tumor cannot be obtained because we did complete ablation of the local tumor which was in line with the clinical practice, different from previous reported partial ablation research (Nat Commun, 13(1):6203, PMID: 36261437; J Transl Med, 20(1):433, PMID: 36180876; Adv Sci, 9(7):e2102182, PMID: 35037422). In our previous study (Cell Mol Immunol, 18(9):2153-2164, PMID: 32385362), we also wanted to investigate the local T-cell infiltration but failed because of complete ablation. IHC staining of CD3 on day 7 after MWA also validated that there were no infiltrated T cells in the ablated tumor (figure below). Moreover, 25 days after MWA, the ablated tumor has already been absorbed.

In our study, we focused on the long-term therapeutic effect after the complete ablation of the unifocal primary tumor. However, MMTV-PyMT mice grow multiple tumors at the same time, and we cannot accurately discern every tumor focus, which makes it difficult to completely ablate the primary tumors. To make our conclusion more convincing, we used a breast cancer cell line Py8119 derived from MMTV-PyMT tumor. Similar results were obtained in the Py8119 tumor model.

Figure legend: WT C57BL/6 mice were implanted with Py8119 mammary tumor cells on day 0. Mice were treated with MWA on day 10. The ablated tumors were collected on 0D, 3D, and 7D after MWA. Representative H-E staining (a-c) or IHC staining (c) sections were shown. Scale bar, 20um.

2. 2DG (2-deoxy-D-glucose) is a glycolysis inhibitor that preferentially targets tumor cells. However, the authors showed that 2DG had no effect on tumor cells (e.g., Fig. 4B), and only affects T cells, which raises doubts. Previous studies have shown that T cells express high levels of MDR1, which may pump the drug out and facilitate tumor cells to take up the drug.

Response: We are sorry that we didn't make this important experiment design clear. The rechallenge experiments were performed on day 35 (5 days after the last injection of 2DG) to exclude the direct anti-tumor effect of 2DG. Theoretically, there was no remaining 2DG on day 35 because the plasma half-life of 2DG was only 48min (Int J Mol Sci, 21(1):234, PMID: 31905745). UPLC-ESI-MS/MS was used to further validate that there was no remaining 2DG 5 days after the last injection of 2DG (fig 2d). (Page 10: line 187-188)

3. The mechanistic experiments are insufficient. The inclusion of 2DG possibly accelerates apoptosis of memory CD8+ T cells. It is crucial for the authors to address this concern and carefully evaluate the potential impact of 2DG on memory T cell viability in their experiments. What is the impact of 2DG on fatty acid metabolism, glutaminolysis, glycogen metabolism and other metabolic pathways upon glycolysis inhibition, which should be

analyzed.

Response: We appreciate the comments. According to your suggestion, we evaluated the apoptosis of CD44^{hi}CD62L⁺CD8⁺T cells in the spleen on day 35 between the MWA+2DG and MWA+PBS groups. No difference was found between the two groups (fig 3g). Moreover, CD8⁺T cells were isolated from spleens of mice in the MWA+2DG and MWA+PBS groups for RNA-seq to evaluate other metabolism pathways (fig 6k). The results demonstrated that the scores of fatty acid oxidation and glutaminolysis process of CD8⁺T cells decreased after MWA+2DG compared with the MWA+PBS (fig 6k). No difference was observed in the score of the glycogen catabolic process of CD8⁺T cells between the MWA+2DG and MWA+PBS groups (fig 6k). (Page 28-29: line 508-518)

4. There are serious concerns with the analysis of memory T cells presented in Figure 3. Analyzing memory T cells should be specific, such as using tetramers or CD45.1-OT1 to distinguish between specific and bystander memory T cells, rather than analyzing all T cells collectively. The analysis should be more targeted and focused to accurately evaluate and understand the presence and function of memory T cells. It is important for the authors to address these concerns.

Response: Thanks for your helpful advice. AH1-tetramer has been used in our study to exclude bystander T cells in the 4T1 tumor model. In tumor-specific T cells, similar results were obtained (fig 4g). Thanks sincerely for your advice which helped us a lot. (Page 19-20: line 355-362)

5. For Figure 7, the authors should use tetramers to analyze tumor-specific T cells. For patients, infections can significantly influence the experimental results. Therefore, there are concerns regarding this part of the results.

Response: Thanks for your comments. Firstly, blood samples were collected at very short intervals. Thus, other factors such as infections could be excluded. Secondly, malignant Liver and breast tumors are extraordinarily heterogeneous diseases among the tumors that have so far been identified (Cancer Lett, 379(2):191-7, PMID: 26213370; J Clin Invest, 121(10):3786-8, PMID: 21965334). There aren't acknowledged tumor antigens for all patients with breast or liver tumors, which results in the lack of tetramers to accurately detect tumor-specific T cells in all patients with breast or liver tumors. CD39 is capable of discriminating tumor-reactive T cell clones from bystander clones such as virus-specific T cells from the tumor or periphery (Nat Commun, 13(1):1935, PMID: 35410325; Nature, 557(7706):575-579, PMID: 29769722; Cell Res, 32(6):530-542, PMID: 35165422). CD39 was utilized to detect tumor-specific T-cell clones in our ex-vivo experiments. Similar results were observed in our previous cohorts (fig 7c, and 7g). To make our results more convincing, 9 consecutive patients with early-stage breast cancer who received MWA in our phase II clinical trial (NCT04805736) were enrolled and similar results were obtained (fig 7d). The sample size of our previous cohorts (fig 7c, and 7g) decreased to 3 because of the lack of preserved PBMCs. Thanks again for your concerns which improved our manuscript a lot. (Page 32-33: line 563-582)

REVIEWER COMMENTS

Reviewer #1 (Remarks to the Author):

In this revised version, the authors have included several new experiments. However, those only addressed part of my concerns.

1) The main hypothesis is still not adequately supported by data, and the mechanism of memory induction by 2-DG remains elusive. The authors claim that a glycolytic metabolic bias in T cells after MWA in breast cancer limits their memory differentiation. They then move into mouse models without characterizing the metabolic phenotype of T cells over the course of MWA or 2-DG treatment. Instead, they perform RNA sequencing on CD8 T cells, about 25 days following the first administration of 2-DG. The observed differences in metabolic gene signatures might be secondary to the memory differentiation. The authors then identify STAT1 as a potential transcription factor mediating the memory induction by 2DG. However, the in vitro experiment is not convincing: the memory induction by 2-DG is negligible and a control condition of fludarabine alone was not included.

Finally, systemic 2-DG treatment can have large systemic effects or affect the phenotype of other immune cells, which can all impact memory T cell development.

2) The memory signature in the human breast cancer samples remain confusing. Why did the authors apply a murine memory CD8 T cell signature to their human samples? Furthermore, the expression data of KLRG1 creates confusion, because it is highly upregulated in both the memory subset (GZMK T cells) and the non-memory subset (GZMH T cells).

3) The number of independent repeats were not included in the figure legends nor methods, unlike what the authors claimed in the rebuttal letter.

Reviewer #2 (Remarks to the Author):

In this manuscript by Tang X. et al., the authors answered well to the concerns I raised. Yet, two essential points were not adequately addressed.

1. While I understand the difficulty to explore antigen release after MWA, this effect must be a key advantage of this combination treatment. DC accumulation and enhanced T cell responses support the antigen release after the MWA treatment. However, the novelty and strong point of this study must be clarifying the style of antigen release after the MWA treatment, which strongly induced tumor antigen-specific T cell responses. Therefore, the current explanation and data are not sufficient to support the novelty and strong point of this study.

2. Another important point is the increase of CD8+ central-memory T cells, which contributed to the long-lasting antitumor effect induced by the combination treatment. The authors did not perform sc-RNA-seq analysis to show the differentiation of tumor antigen-specific T cells due to various reasons and use traditional methodology to delineate central memory T cell subsets after the combination treatment. Considering the essential contribution of CD8+ central-memory T cells to the long-lasting antitumor effect, which the authors argue as a key issue of this study, the differentiation of tumor antigen-specific CD8+ T cells should be examined. For example, as the authors have shown the increase of tetramer+ CD8+ T cells, based on the TCR usage of these tetramer+ CD8+ T cells, tumor antigen-specific CD8+ T cells can be explored with sc-RNA-seq analysis.

Reviewer #3 (Remarks to the Author):

The authors have appropriately addressed my concerns raised during the last round.

Reviewer #1 (Remarks to the Author):

In this revised version, the authors have included several new experiments. However, those only addressed part of my concerns.

1) The main hypothesis is still not adequately supported by data, and the mechanism of memory induction by 2-DG remains elusive. The authors claim that a glycolytic metabolic bias in T cells after MWA in breast cancer limits their memory differentiation. They then move into mouse models without characterizing the metabolic phenotype of T cells over the course of MWA or 2-DG treatment. Instead, they perform RNA sequencing on CD8 T cells, about 25 days following the first administration of 2-DG. The observed differences in metabolic gene signatures might be secondary to the memory differentiation. The authors then identify STAT1 as a potential transcription factor mediating the memory induction by 2DG. However, the in vitro experiment is not convincing: the memory induction by 2-DG is negligible and a control condition of fludarabine alone was not included.

Finally, systemic 2-DG treatment can have large systemic effects or affect the phenotype of other immune cells, which can all impact memory T cell development.

Response: Thanks for your concerns. We agreed that it was necessary to characterize the metabolic phenotype (Immunity, 54(4):829-844.e5, PMID: 33705706; Nat Rev Immunol, 21(11):718-738, PMID: 33981085) of T cells over the course of the treatment. To reach this purpose, we isolated splenic CD8+T cells on day 15 (5 days following the first administration of 2DG). The results indicated that glycolysis was inhibited while OXPHOS was not throughout the combination therapy. Consistent with our hypothesis, CD8+T cells relied more on OXPHOS than on glycolysis throughout the combination therapy compared to the MWA+PBS group (fig S6d). The results also indicated that other metabolic changes of CD8+T cells after the combination therapy observed on day 35 might not be secondary to memory formation because there were significant differences in other metabolic pathways between the MWA+PBS and MWA+2DG groups throughout memory formation (fig S6d). We have modified the description of the results in our manuscript. (Page 31, line 558-565)

To make our result more convincing, we re-analyzed our data of RNA-seq of isolated CD8+T cells. The results indicated that on day 35 (25 days after MWA), the expression of Stat1 was also elevated in the MWA+2DG group compared to the MWA+PBS group, which was consistent with our previous estimated results. According to your suggestions, we repeated our in-vitro experiment of STAT1 inhibitor with a control group, and similar significant results were obtained (fig 6p and S6e). We have added them to the results part. (Page 32, line 570-572)

To clarify whether the influence of 2DG on the other immune cells contributed to the memory phenotype of CD8+T cells, additional experiments were performed. Splenic immune cells of donor mice treated with MWA+2DG or MWA+PBS as above were collected 5h after the injection of 2DG or PBS every two days. The isolated immune cells with CD8+T cells depleted were immediately transferred into the recipient mice treated with MWA+2DG or MWA+PBS (fig 4h). The percentage and absolute number of tumor-specific

CD44^{hi}CD62L⁺CD8⁺T cells were not increased after the transfer of 2DG-treated non-CD8 immune cells compared to the transfer of PBS-treated non-CD8 immune cells in the mice treated with MWA+PBS (fig 4i). The results indicated that other immune cells had no significant impact on memory T cell development (fig 4h-i). We have added them to the results part. (Page 20, line 373-382)

We thank you sincerely for making our study better.

2) The memory signature in the human breast cancer samples remain confusing. Why did the authors apply a murine memory CD8 T cell signature to their human samples? Furthermore, the expression data of KLRG1 creates confusion, because it is highly upregulated in both the memory subset (GZMK T cells) and the non-memory subset (GZMH T cells).

Response: Thanks for your concerns. We are so sorry that we didn't make the signature used in our manuscript clear. Although the memory CD8 T cell signature used in our study was originally studied in mice (Proc Natl Acad Sci U S A, 101(48):16885-90, PMID:15548615), the signature used in our human samples was obtained from the human collections (collections of human gene sets) in the MsigDB dataset (GOLDRATH_IMMUNE_MEMORY, ID: M10845) contributed by Kate Stafford (MSigDB Team). To make our conclusion more convincing (memory signature score increased in memory subset GZMK T cells), one more memory signature was used in the memory subset to validate our results (Science, 374(6574):abe6474, PMID: 34914499). Similar results were obtained (fig S1c). We have added it to the results part. (Page 6, line 122)

We apologize that we didn't explain the expression of KLRG1 clearly in our study. The memory phenotype of CD8⁺T cells has slightly increased after MWA. However, the expression of KLRG1, low expression of which is parallel to memory phenotype (Immunity, 48(4):716-729.e8, PMID: 29625895), was upregulated in two clusters of CD8⁺T cells (fig S1d-e) after MWA. The results indicated that there might be factors that were unfavorable for memory generation leading to only a slight increase of the memory phenotype of CD8⁺T observed after MWA. Cellular metabolism status can determine the fate of CD8⁺T cells. Thus, metabolism features were then analyzed in our study. We have modified the description to make our manuscript easier to understand. (Page 6, line 122-131)

Thanks again for improving our manuscript.

3) The number of independent repeats were not included in the figure legends nor methods, unlike what the authors claimed in the rebuttal letter.

Response: We are sorry that we forgot to add the statement in the figure legends. We have added the number of independent repeats in our manuscript.

Reviewer #2 (Remarks to the Author):

In this manuscript by Tang X. et al., the authors answered well to the concerns I raised. Yet, two essential points were not adequately addressed.

1. While I understand the difficulty to explore antigen release after MWA, this effect must

be a key advantage of this combination treatment. DC accumulation and enhanced T cell responses support the antigen release after the MWA treatment. However, the novelty and strong point of this study must be clarifying the style of antigen release after the MWA treatment, which strongly induced tumor antigen-specific T cell responses. Therefore, the current explanation and data are not sufficient to support the novelty and strong point of this study.

Response: We appreciate your concerns. Tumor antigens that are released after necrosis induced by thermal ablation drain to nearby lymph nodes, where they can stimulate immature DCs and naive T cells (Nat Rev Cancer; 14(3):199-208; PMID: 24561446). As the high temperature induced by MWA (60-80 °C), tumor cells were immediately killed with necrosis instead of apoptosis, which could induce antigen release. Thus, it is important to distinguish necrosis and apoptosis of tumor cells after MWA to clarify the style of antigen release after MWA.

One of the biggest differences between necrosis and apoptosis is the integrity of cell membrane. To simulate MWA, we performed in-vitro thermal treatment on 4T1-zsgreen cells cultured in dishes. Before the thermal treatment, the culture medium of 4T1-zsgreen was replaced with PBS. The dishes were sealed with parafilm and submerged in a water bath for 3 mins at 60 °C or 80 °C, and 37 °C was set as the control group. Much more zsgreen released out of the cells heated at 60 °C or 80 °C compared to 37 °C was found, supporting the ruptured cell membrane induced by necrosis of tumor cells after MWA (fig S4e). To make the necrosis induced by MWA more convincing, 4T1 tumors were collected after MWA 5h for transmission electron microscope (TEM). The necrosis of the cells after MWA was validated by the cell morphology (fig S4f). Moreover, in-vivo experiments were performed to further validate the antigen release induced by MWA. 4T1-zsgreen tumors 5h after MWA or non-treatment were collected for detection of released intracellular zsgreen in the tissue interstitial fluid. More zsgreen was detected from the tissue interstitial fluid of mice in the MWA group compared to the non-treatment group. Thus, MWA induced tumor antigen release by the necrosis of tumor cells.

However, it is necessary to determine whether the released zsgreen could be drained into tumor-draining lymph nodes and be captured by antigen-presenting cells. We then harvested the tumor-draining lymph nodes of mice in the MWA and the non-treatment groups (5h after MWA). Consistently, more zsgreen was observed from the tissue interstitial fluid of tumor-draining lymph nodes after MWA compared to non-treatment (fig S4e). Moreover, more zsgreen+DCs but not zsgreen+macrophages were observed in the tumor-draining lymph node of the MWA group compared to the non-treatment group (fig S4e). The above results indicated that the released tumor antigen could be drained into tumor-draining lymph nodes and be captured by DCs for downstream tumor antigen-specific T-cell responses. The result part has been modified based on the additional data (Page 20, line 366-372). We appreciate that your concern helped us improve our study a lot.

Figure legend: Graphic abstract of the style of antigen release after MWA.

2. Another important point is the increase of CD8⁺ central-memory T cells, which contributed to the long-lasting antitumor effect induced by the combination treatment. The authors did not perform sc-RNA-seq analysis to show the differentiation of tumor antigen-specific T cells due to various reasons and use traditional methodology to delineate central memory T cell subsets after the combination treatment. Considering the essential contribution of CD8⁺ central-memory T cells to the long-lasting antitumor effect, which the authors argue as a key issue of this study, the differentiation of tumor antigen-specific CD8⁺ T cells should be examined. For example, as the authors have shown the increase of tetramer⁺ CD8⁺ T cells, based on the TCR usage of these tetramer⁺ CD8⁺ T cells, tumor antigen-specific CD8⁺ T cells can be explored with sc-RNA-seq analysis.

Responses: Thanks for your concerns. According to your suggestions, CD8⁺T cells in the peripheral blood were collected from mice treated with MWA+PBS and MWA+2DG on day 35. CD8⁺T cells were incubated with oligo-barcoded AH1-specific tetramer. Then, single-cell RNA seq and single-cell TCR were performed on these CD8⁺T cells. In shared TCR clonotypes of CD8⁺T cells between the two groups, tetramer-positive clonotypes of CD8⁺T cells were determined as tumor-specific CD8⁺T cells for further analysis (Fig 6k). By pseudo-time analysis, the trajectory of tumor-specific CD8⁺T cell differentiation was investigated. The results indicated that the differentiation of tumor-specific CD8⁺T cells was from CD8⁺Tnaive to early activated CD8⁺T, and then part of early activated CD8⁺T differentiated into CD8⁺Tcm, which was promoted by MWA+2DG. Moreover, CD8⁺Tcm

and the left early activated CD8+T could further differentiate into stem-like CD8+Tpx and finally differentiate into CD8+Teff (fig 6m and S6c). We have added the data to the results part (Page 30, line 530-542). We thank you again for your concern to improve our study.

Figure legend: Graphic abstract of the trajectory of tumor-specific CD8+T cell differentiation.

Reviewer #3 (Remarks to the Author):

The authors have appropriately addressed my concerns raised during the last round.

REVIEWER COMMENTS

Reviewer #1 (Remarks to the Author):

Tang et al performed additional experiments, but those could not address my concerns about the proposed metabolic in vivo mechanism:

1) The authors now investigated metabolic gene expression 15 days after MWA. It is however still not clear if the mouse model reflects their human observation: is there also a glycolytic bias when comparing mouse CD8 T cells before and after MWA?

2) The experiment in which the authors deplete CD8 T cells from the immune cells sorted from spleens of mice treated with MWA + 2DG, followed by retransfer does not exclude the possibility of systemic CD8-extrinsic effects of 2DG on memory differentiation. Only CD8-specific genetic interference with the proposed mechanistic signaling cascade (e.g. STAT1 KO T cells) would provide strong in vivo evidence of the hypothesis.

3) The KLRG1 data remains confusing. Which criteria do the authors use to define low versus high KLRG1 expression. Are the KLRG1-expressing cells also positive for IL7R, which is the supposed subset able to down-regulate KLRG1 and contribute to the long-lived memory pool?

Reviewer #2 (Remarks to the Author):

The paper by Zhou W. et al. entitled ""Glycolysis inhibition potentiates local tumor ablation-induced anti-tumor immunity via CD44^{hi}CD62L⁺CD8⁺T cells in a TDLN-dependent manner"" has been revised extensively, and I have no further concern for publishing this paper.

Tang et al performed additional experiments, but those could not address my concerns about the proposed metabolic in vivo mechanism:

1) The authors now investigated metabolic gene expression 15 days after MWA. It is however still not clear if the mouse model reflects their human observation: is there also a glycolytic bias when comparing mouse CD8 T cells before and after MWA?

Response:

Thanks for your concern. We agree that it is important to determine the reasonability of our mouse models. Additional experiments were performed to evaluate the metabolic status of peripheral CD8+T cells before and after MWA in our mouse model (Fig R1a-b). We previously found a glycolysis bias in human CD8+T cells after MWA based on the decreased score of OXPHOS and the unchanged score of glycolysis. The relative expression of metabolic genes related to OXPHOS and glycolysis were evaluated. The results indicated that the glycolysis didn't change but OXPHOS decreased after MWA in peripheral CD8+T cells from our mouse model (Fig R1a-b), which was consistent with our human observation.

We have added the following to the result section: "Consistent with our human observation, the glycolysis bias of CD8+T cells in the mouse model after MWA was confirmed." (line170-171)

Thanks again for your helpful concerns.

Fig R1: (a-b) Relative expression of metabolic genes in CD8+T cells isolated from the spleens of mice 0 days before and 7 days after MWA (n = 5).

2) The experiment in which the authors deplete CD8 T cells from the immune cells sorted from spleens of mice treated with MWA + 2DG, followed by retransfer does not exclude the possibility of systemic CD8-extrinsic effects of 2DG on memory differentiation. Only CD8-specific genetic interference with the proposed mechanistic signaling cascade (e.g. STAT1 KO T cells) would provide strong in vivo evidence of the hypothesis.

Response:

Thanks for your concern. We chose Stat1-/- mice to replicate our experiments. Stat1-/- mice could hardly survive because of the lack of STAT1 although they were maintained under specific pathogen-free conditions. According to your advice, CD8-specific genetic interference was performed in vivo to make our results more convincing. AAV-ark313-DIO-

Editorial Note: Figure below created with BioRender.com and released under a Creative Commons Attribution-NonCommercial-NoDerivs 4.0 International license.

shStat1-Zsgreen or AAV-ark313-DIO-shNC-Zsgreen (Cell, 186(2):446-460.e19, PMID: 36638795) were injected into Cd8a-Cre mice (Fig R2a). By the double-floxed inverted orientation (DIO) system, the shRNA target at Stat1 and Zsgreen only express in cre+ cells (CD8a+ cells in our mouse model) but not other cells. Importantly, the percentage of CD44^{hi}CD62L+ subset decreased with the in-vivo knockdown of STAT1 in CD8+T cells from the MWA+2DG group (Fig R2b-c) but not in the MWA+PBS group. The results provide in-vivo evidence of the CD8-intrinsic effects of 2DG on memory differentiation.

We have added the following to the result section: **“To make our results more convincing, CD8-specific genetic interference of STAT1 was performed in vivo by CD8a-Cre mice and AAV-ark313 with the double-floxed inverted orientation (DIO) system (Fig 6q-r). The shRNA target at Stat1 and Zsgreen only express in Cre+ cells (CD8a+ cells in our mouse model) but not in other cells. The percentages of CD44^{hi}CD62L+ subset decreased with the in-vivo knockdown of STAT1 in CD8+T cells from the MWA+2DG group (Fig 6s).”** (line572-578)

We thank you sincerely for your advice which makes our manuscript more convincing.

Fig R2: a Experimental design for **(b-c)**. CD8a-Cre C57BL/6 mice were implanted with Py8119 mammary tumor cells and injected with AAV-CMV-DIO-shStat1/shNC-Zsgreen intravenously on day 0. Mice were treated with MWA on day 10 and were intraperitoneally injected with 2DG or PBS every two days from day 10 to day 30. Spleens were collected on day 35 for flow cytometry. **b** Knockdown of total STAT1 of Zsgreen+CD8+T cells in spleens of CD8a-cre mice injected with AAV (n = 3 per group). **c** Percentage of CD44^{hi}CD62L+Zsgreen+CD8+T cells in spleens of mice from the 6 groups in **(a)** on day 35 (n = 3 per group).

3) The KLRG1 data remains confusing. Which criteria do the authors use to define low versus high KLRG1 expression. Are the KLRG1-expressing cells also positive for IL7R, which is the supposed subset able to down-regulate KLRG1 and contribute to the long-lived memory pool?

Response:

Thanks. We defined KLRG1+CD8+T cells as high KLRG1 expression while KLRG1-CD8+T cells as low KLRG1 expression (Fig R3a). We found that only a part of the KLRG1-expressing cells were also positive for IL7R (Fig R3b). Thus, KLRG1 was analyzed in a pool of memory and non-memory CD8+T cells previously. We agreed with you that the analysis of KLRG1 should be performed in the memory CD8+T cells (IL7R+) subset to accurately evaluate the memory phenotype after MWA. In IL7R+CD8+T cells, the expression of KLRG1 showed no significant difference after MWA (Fig R3c). But in IL7R-

CD8+T cells, the expression of KLRG1 increased after MWA (Fig R3d). The results indicated that the non-memory CD8+T cells obtained the phenotype of terminal differentiation but not memory CD8+T cells. According to your concern, our previous results of KLRG1 might confuse readers and we decided to delete our previous results of KLRG1. The deletion of results of KLRG1 would not affect our previous conclusion that the memory phenotype of CD8+T cells slightly increased after MWA.

Thanks again for making our manuscript better.

Fig R3: **a** Ridgeline Plot of the expression of KLRG1 in two clusters of CD8+T cells. **b** Feature plots of IL7R gene expression in the KLRG1+CD8+T cells. **c-d** Violin plots of expression of KLRG1 in IL7R+CD8+T or IL7R-CD8+T cells before and after MWA.

REVIEWERS' COMMENTS

Reviewer #1 (Remarks to the Author):

In this round of revision, Tang et al have addressed my final concerns. I believe the manuscript may be published.

2 final points:

1) I agree with not showing the KLRG1 data, this will not alter the interpretation of the memory status of T cells post MWA. However, KLRG1 is still mentioned in the methods, including a reference. I suggest a more thorough and careful editing of the manuscript before publication.

2) The CD8-specific deletion with sh-expressing AAV is a big plus to the STAT-mediated memory induction. I think it would be very informative to the reader to include the percentage of successful transduction in the total CD8 T cells as supplementary data (percentage of Zsgreen CD8 T cells in the total CD8 T cells in the spleen at day 35).

Reviewer #1 (Remarks to the Author):

In this round of revision, Tang et al have addressed my final concerns. I believe the manuscript may be published.

2 final points:

1) I agree with not showing the KLRG1 data, this will not alter the interpretation of the memory status of T cells post MWA. However, KLRG1 is still mentioned in the methods, including a reference. I suggest a more thorough and careful editing of the manuscript before publication.

Response: Thanks for your concern. We have edited our manuscript carefully.

2) The CD8-specific deletion with sh-expressing AAV is a big plus to the STAT-mediated memory induction. I think it would be very informative to the reader to include the percentage of successful transduction in the total CD8 T cells as supplementary data (percentage of Zsgreen CD8 T cells in the total CD8 T cells in the spleen at day 35).

Response: Thanks for your concern. We have added the transduction efficiency in our supplementary data (Fig S6f).